# SUBSAMPLING IN LARGE GRAPHS USING RICCI CURVATURE

**Shushan Wu, Huimin Cheng, Jiazhang Cai, Ping Ma & Wenxuan Zhong**
Department of Statistics
University of Georgia
Athens, GA 30605, USA
`{Shushan.Wu,huimincheng,jc27544,pingma,wenxuan}@uga.edu`

## ABSTRACT

In recent decades, an increasing number of large graphs with millions of nodes have been collected and constructed. Despite their utility, analyzing such graphs is hindered by high computational costs and visualization difficulties. To tackle these issues, researchers have developed graph subsampling approaches that extract representative nodes to provide a rough sketch of the graph that preserves its global properties. By selecting representative nodes, these graph subsampling methods can help researchers estimate the graph statistics, e.g., the number of communities, of the large graph from the subsample. However, the available subsampling methods, such as degree node sampler and random walk sampler, tend to overlook minority communities, as they prioritize nodes with high degrees. To address this limitation, we propose leveraging community information hidden within the graph for subsampling. Although the community structure is typically unknown, geometric methods can reveal community structure information by defining an analog of Ricci curvature on the graph, known as the Ollivier Ricci curvature. We introduce a new subsampling algorithm called the Ollivier Ricci curvature Gradient-based subsampling (ORG-sub) algorithm based on our asymptotic results regarding the within-community and between-communities edges' OR curvature. The ORG-sub algorithm makes two significant contributions: Firstly, it provides a rigorous theoretical guarantee that the probability of taking all communities into the final subgraph converges to one. Secondly, extensive experiments on synthetic and benchmark datasets demonstrate the advantages of our algorithm.

## 1 INTRODUCTION

As we enter the big data era, our capacity to access large graphs (a.k.a. networks) has provided unprecedented opportunities and challenges. For example, Wang et al. (2011) presented a Twitter social network, which has more than 190 million nodes (users) who generate more than 65 million edges (tweets) every day (Wang et al., 2011). Such huge networks enable researchers to tackle more complex problems. However, they pose great challenges to storing, visualizing, and analyzing since their sheer volumes render many computational methods infeasible.

**Graph subsampling.** Graph subsampling is a commonly used technique to address this scalability issue because of its simplicity and efficiency. Graph subsampling aims to take a subgraph that preserves critical features of the full graph. Various graph subsampling methods that preserve different graph properties have been proposed, including node sampling (Mall et al., 2013; Zeng et al., 2019), edge sampling (Krishnamurthy et al., 2005), and exploration sampling (Goodman, 1961; Leskovec et al., 2005; Hübler et al., 2008; Maiya & Berger-Wolf, 2010). Researchers evaluate the graph subsampling approaches by measuring the similarity between the features of the original graph and those of the subgraph. The features include, e.g., degree distribution (Adamic et al., 2001), minimum cut (Hu & Lau, 2013), and the number of triangles (Seshadhri et al., 2014).

An important graph feature that has received less attention is the number of communities (denoted by $M$), which plays a crucial role in identifying the community structures. Network data often have natural communities, and the identification of these communities helps answer vital questions in a

variety of fields (Rohe et al., 2011). For example, communities in social networks may represent groups of people who share a similar interest, and communities in protein-protein interaction networks could be regulatory modules of interacting proteins (Rohe et al., 2011). In this paper, we focus on the setting where $M$ is a fixed model parameter, and there is a ground truth about $M$. This setting is widely used in many models, e.g., stochastic block model (SBM) (Holland et al., 1983) and its variants such as degree-corrected SBM (DCBM) (Karrer & Newman, 2011). The SBM family is arguably the most widely-used generative model for community detection from a theoretical perspective. In fact, Vaca-Ramírez & Peixoto (2022) performed a systematic analysis of the quality of fit of the SBM for 275 real networks and observed that "SBM is capable of providing an accurate description for the majority of networks considered". Indeed, there are other settings where the number of communities is not fixed. For example, Olhede & Wolfe (2014) used SBM to approximate a nonparametric graphon model, under which case the number of communities is a hyperparameter and is not fixed. The latter hyperparameter case is beyond the scope of this paper.

Under the setting where $M$ is a model parameter, many community detection methods have been proposed, such as modularity maximization (Newman, 2006; Good et al., 2010), spectral clustering (Von Luxburg, 2007; Rohe et al., 2011; Liu et al., 2018), and pseudo-likelihood based methods (Amini et al., 2013; Wang et al., 2021). The theoretical properties of most community detection methods, such as consistency and asymptotic distributions, are built based on the assumption that $M$ is known (Ma et al., 2021). In addition, $M$ is usually required as an input for those community detection algorithms. However, in practice, we do not have the information of $M$, which significantly diminishes the usefulness of the aforementioned methods. Existing methods for estimating $M$ are usually very expensive. For example, the cross-validation method proposed by Li et al. (2020) requires a computational cost that is cubic in the number of nodes $n$. When there are thousands or millions of nodes, the computational cost is unaffordable.

Thus, it is highly desirable for subsampling methods to yield subgraphs with $\tilde{n} << n$ nodes preserving the number of communities $M$, such that we can use it to get an accurate estimation while reducing the computational cost. Despite many successful applications, existing subsampling methods tend to leave out minority communities (i.e., a community with a smaller number of nodes) because nodes with high degrees are more likely to be sampled into subgraphs. Consequently, these subsampling methods usually underestimate $M$, especially for graphs with imbalanced community structures.

To overcome the shortcomings of existing methods, we develop a graph subsampling method that yields subgraphs that can be used to accurately estimate $M$. Achieving this goal is challenging since the community structure is hidden and unavailable. Fortunately, recent studies indicate that the community structure is a geometric phenomenon by considering a graph as a Riemannian geometric object (Ni et al., 2015). Some insights into the community information can be obtained by applying some geometric methods to a graph (Ni et al., 2015; 2019; Sia et al., 2019).

**Ollivier Ricci Curvature of Graph.** In particular, a graph can be regarded as a discrete version of a Riemannian manifold (Ni et al., 2019). A node of the graph is analogous to a point on a Riemannian manifold, and a pair of connected nodes in a graph is analogous to two points connected by a geodesic on a manifold. The partition of a graph into communities is analogous to the geometric decomposition of a Riemannian manifold (Ni et al., 2019). Ricci curvature is a key tool for the geometric decomposition of a Riemannian manifold. It measures how the Riemannian manifold deviates from the flat manifold. Recently, Lin et al. (2011) defines an analog of Ricci curvature for the graph, i.e., the Ollivier Ricci (abbreviated as OR) curvature. Previous empirical results have shown the OR curvature is related to the connectivity of the graphs (Ni et al., 2015; Gosztolai & Arnaudon, 2021). However, the relationship between OR curvature and connectivity is insufficiently explored in theoretical evidence. In this paper, we theoretically show that the OR curvatures of edges within a densely connected community are asymptotically larger than those of edges between two sparsely connected communities, as the size of the graph increases.

**Ollivier Ricci Curvature Gradient Based Graph Subsampling.** Based on our theoretical result, we propose an OR curvature gradient-based graph subsampling algorithm (abbreviated as ORG-sub). Specifically, ORG-sub randomly chooses one edge as the starting point of the subgraph and calculates the OR curvature of the selected edge. ORG-sub then gradually expands the subgraph by taking the next edge whose OR curvature shows the largest difference from the OR curvature of the previously taken edge. Here, we define the difference as the OR curvature gradient (ORG). We use the ORG to guide the expansion of ORG-sub, i.e., direct ORG-sub which edge to take next. All edges that

ORG-sub took expand into the final subgraph. Our proposed ORG-sub enjoys two main advantages. First, ORG-sub has a rigorous theoretical guarantee. In particular, under the SBM scenario, we prove that the probability of ORG-sub taking all communities into the final subgraph converges to one faster than the random walk algorithm, indicating that even nodes in the minority community are subsampled. More importantly, we theoretically show that the estimation of $M$ by the subsampled graph converges to $M$ of the full graph. Second, our extensive empirical experiments based on the simulated and real-world datasets show that the estimator based on ORG-sub subgraphs accurately estimates $M$ while greatly reducing the computation cost.

## 2 PRELIMINARIES

### 2.1 NOTATIONS AND DEFINITIONS

A graph is made up of nodes that are connected by edges. Considering that directed graphs don't approximate Riemannian manifolds in discrete form since geodesics are always bidirectional, we only focus on undirected graphs in this paper. Denote a graph as $G = \langle V, E \rangle$, where $V$ is the node set and $E$ is the edge set. Operator $|\cdot|$ calculates the cardinality of the set. We then denote the cardinality of the node set in a graph by $|V|$ ($|V| = n$), the number of edges in a graph by $|E|$, and the neighborhood set of a node $v$ by $\delta(v) = \{w|(v, w) \in E\}$. The degree of a node $v$ is defined as the cardinality of its neighborhood: $d_v = |\delta(v)|$. For a subset of $V$, $S \subseteq V$, the subgraph induced by the subset $S$ is denoted by $G[S] = (S, E_S)$, where the edge set $E_S = \{(v_S, w_S) \in E | v_S \in S, w_S \in S\}$. The neighborhood of the node set $S$ is defined by $N(S) = \bigcup_{v \in S} \delta(v)$, and the neighboring edge set of edge $e = (u, v)$ is $\Delta(e) = \{(x, y)|x \in \{u, v\}, y \in \delta(x) \setminus \{u, v\}\}$. Let $\widehat{M}(G[S])$ denote the estimator of $M$, obtained by using the subgraph $G[S]$ for estimation.

### 2.2 OLLIVIER RICCI CURVATURE OF GRAPHS

A Riemannian manifold $(\mathcal{M}, g)$ is defined as a smooth manifold $\mathcal{M}$ equipped with a metric $g : \mathcal{TM} \times \mathcal{TM} \rightarrow \mathbb{R}$, where $\mathcal{TM}$ is the tangent space of $\mathcal{M}$ (Lee, 2018). Ricci curvature of Riemannian manifold measures the deviation of the manifold from a flat manifold (Lee, 2006; Ni et al., 2015; Samal et al., 2018). The "flat" means the distance between two nodes is the same as the distance between the two local spaces centered around the nodes. In the discrete version, the space defined on the node is discretely uniformly distributed in the node's neighborhood, then the distance between two spaces can be explained as the average distance between the neighborhoods of the two nodes. Through such an analogy, the counterpart of Ricci curvature, Ollivier Ricci curvature (Lin et al., 2011), is defined in graphs. Previous empirical results show that

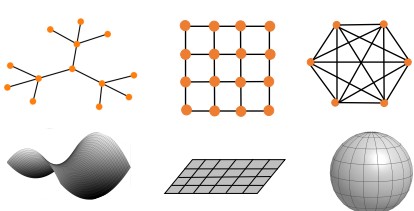

Figure 1: The left one is a tree-structured graph corresponding to a hyperbolic manifold with negative curvature. The middle one is a grid-structured graph corresponding to a Euclidean space with zero curvature. The right one is a complete graph $K_6$, which is embedded in a spherical surface with positive curvature.

the OR curvature of graphs is related to the Ricci curvature of the underlying manifold: the graph with large (or small) OR curvature corresponds to the underlying manifold with large (or small) Ricci curvature (Ni et al., 2015; Samal et al., 2018; Ni et al., 2019; Gosztolai & Arnaudon, 2021). Theoretical results have shown the OR curvature of random geometric graphs converges to the Ricci curvature of the underlying Riemannian manifold (Boguna et al., 2021; van der Hoorn et al., 2021). Here we use a toy example to illustrate the relationship between the OR curvature of graphs and the corresponding underlying manifold. In Figure 1, the hyperbolic space (with negative curvature) is especially suited for tree-structured graphs (Nickel & Kiela, 2017), the Euclidean space (with zero curvature) is especially suited for grid-structured graphs, and the spherical space (with positive curvature) is especially suited for complete graphs (Ni et al., 2015).

We now introduce the definition of OR curvature of the graph. For $\alpha \in [0, 1]$ and any node $u$ with degree $d_u$, we first define the probability distribution of node $u$ by $m_u^\alpha$:

$$m_u^\alpha(x) = \begin{cases} \alpha & \text{if } x = u \\ (1 - \alpha)/d_u & \text{if } x \in \delta(u) \\ 0 & \text{otherwise} \end{cases} \tag{1}$$

Here $\alpha$ is to keep the probability mass of $\alpha$ at node $u$ itself and distribute the rest uniformly over the neighborhood. Following Ni et al. (2018); Ye et al. (2019), we set $\alpha = 0.5$. This means that each node keeps 50% of the probability mass to itself. Let $d(u, v)$ be the geodesic distance between nodes $u$ and $v$, which is the shortest path between two nodes of a graph. We shall present the Wasserstein distance of two nodes in the graph, which is defined as the transportation distance between two nodes' probability distributions $m_u^\alpha$ and $m_v^\alpha$.

**Definition 2.1** (Wasserstein distance). Let $X$ be the metric space with two probability distributions $\mu_1$ and $\mu_2$ with mass 1, respectively. A transportation plan from $\mu_1$ to $\mu_2$ is a mapping $\xi : X \times X \to [0, 1]$ satisfying $\sum_y \xi(x, y) = \mu_1(x)$ and $\sum_x \xi(x, y) = \mu_2(y)$. Substitute $\mu_1$ and $\mu_2$ by probability distributions of nodes $m_u^\alpha$ and $m_v^\alpha$, the Wasserstein distance between two nodes is,

$$W(m_u^\alpha, m_v^\alpha) = \inf_\xi \sum_{u,v \in V} \xi(u,v)d(u,v) \tag{2}$$

The Wasserstein distance $W(m_u^\alpha, m_v^\alpha)$ can be computed by linear programming (Lin et al., 2011).

Then the OR curvature (Ollivier, 2007; Lin et al., 2011) over the edge $(u, v)$ is defined as follows.

**Definition 2.2** (Ollivier Ricci Curvature). Given the Wasserstein distance $W(m_u^\alpha, m_v^\alpha)$ and the geodesic distance $d(u, v)$, we can define the Ollivier Ricci curvature as:

$$\kappa(u, v) = 1 - \frac{W(m_u^\alpha, m_v^\alpha)}{d(u, v)} \tag{3}$$

## 3 OLLIVIER RICCI CURVATURE GRADIENT AND COMMUNITY STRUCTURE

The OR curvature of the graph reveals the properties of the underlying Riemannian manifold and provides insights into the community structure of graphs. Previous work has shown that the large (or small) curvature corresponds to the edge which is more (or less) connected than a grid (Ni et al., 2015; Samal et al., 2018; Ni et al., 2019; Gosztolai & Arnaudon, 2021).

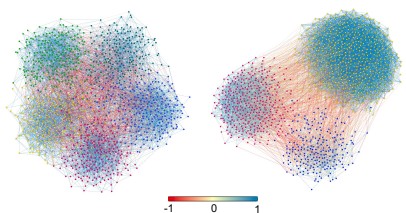

As shown in Figure 2, the left is a graph with five communities, and the right one is a graph with three communities generated by the stochastic block model. The warmer color indicates a smaller curvature, and the colder color indicates a larger curvature. Previous empirical results and the above toy example have related the OR curvature of graphs to the community structure of the graph. However, rigorous theoretical proofs that the OR curvature of the within-community edge is larger than the between-communities edge in a graph are not developed yet. In the Theorem 3.3, we theoretically prove the lower bound of the

Figure 2: The cold-colored edges have large OR curvatures, and the warm-colored edges have small OR curvatures. Nodes with different colors belong to different communities.

within-community edges' OR curvatures are larger than the upper bound of the between-communities edges' OR curvatures in stochastic block models (SBM).

### 3.1 ASYMPTOTIC RESULTS OF OR CURVATURE GRADIENT IN SBM

**Lemma 3.1** (Ollivier-Ricci curvature of random graphs generated by stochastic block models). *Consider a graph with $M$ communities generated from a stochastic block model (SBM) $\mathcal{B}\left( \left\{ p_{in}^{(i)} \right\}_{i=1}^{M}, p_{out}, \{n_i\}_{i=1}^{M} \right)$, where $p_{in}^{(i)}$ represents the probability of an edge connecting nodes in the same community $B_i$, $p_{out}$ represents the probability of an edge connecting nodes from different communities, and $\{n_i\}_{i=1}^{M}$ are the sizes of each community in the SBM. We assume that, for any $i = 1, ..., M$, $p_{in}^{(i)} > p_{out}$, and $\frac{n_i}{n}$ converges to $r_i(1 > r_i > 0)$, as $n$ goes to infinity. Let $n_\vee = \max \{n_i\}_{i=1}^{M}$, and $p_\vee = \max \left\{ p_{in}^{(i)} \right\}_{i=1}^{M}$. We denote the node $x$ from the community $B_i$ by $x_{B_i}$, the edge $(x_{B_i}, y_{B_j})$ where $x$ is from $B_i$ and $y$ is from $B_j$, and the edge's OR curvature by $\kappa(x_{B_i}, y_{B_j})$. The following statements hold for the OR curvature.*

1. *If $p_{out} \leq \sqrt{\frac{(n_i-1)\ln n}{n \ln n_i}}$ and $p_{in}^{(i)} n_i + p_{out}(n - n_i) > \frac{8}{3}\ln n_i$, for the lower bound of within-community OR curvature, we almost surely have*

$$\kappa(x_{B_i}, y_{B_i}) \geq \frac{(n_i-2)p_{in}^{(i)^2} + (n-n_i)p_{out}^2}{(n_i-1)p_{in}^{(i)} + (n-n_i)p_{out}} + \mathcal{O}\Big(\frac{\ln n}{p_{out}^2 n}\Big). \tag{4}$$

2. *If $p_{in}^{(i)^2} n_i + p_{out}^2(n-n_i) > 4\ln n_i$, for the upper bound of within-community OR curvature, we almost surely have*

$$\kappa(x_{B_i}, y_{B_i}) \leq p_{in}^{(i)} + \mathcal{O}\Big(\frac{\ln n_i}{n}\Big). \tag{5}$$

3. *If $p_{out} > \sqrt{(3\ln n_\vee)\Big/ \Big[\frac{p_{in}^{(i)}}{p_{out}}(n_i + n_j) + n - n_i - n_j\Big]}$, for the upper bound of between-communities OR curvature, we almost surely have*

$$\kappa(x_{B_i}, y_{B_j}) \leq \frac{n - n_i - n_j}{n-1}p_{out} + \frac{n_i - n_j - 2}{n-1}p_\vee + \mathcal{O}\Big(\sqrt{\frac{p_\vee \ln n}{p_{out} n}}\Big). \tag{6}$$

**Lemma 3.2.** *Combining the conclusion of statement 1 and statement 2 in Lemma 3.1, almost surely, for any $(x'_{B_i}, y'_{B_i}) \in \Delta((x_{B_i}, y_{B_i}))$, we have*

$$|\kappa(x_{B_i}, y_{B_i}) - \kappa(x'_{B_i}, y'_{B_i})| \leq \frac{p_{out}(p_{in}^{(i)} - p_{out})}{\frac{n_i-1}{n-n_i}p_{in}^{(i)} + p_{out}} + \mathcal{O}\Big(\frac{1}{p_{out} n}\Big). \tag{7}$$

Statements 1 and 2 in Lemma 3.1 together show that under certain conditions, the probability that the OR curvature of within-community edges falls in $\big(\frac{(n_i-2)p_{in}^{(i)^2} + (n-n_i)p_{out}^2}{(n_i-1)p_{in}^{(i)} + (n-n_i)p_{out}}, p_{in}^{(i)}\big)$ goes to 1 as the graph size $n$ goes to infinity. Statement 3 in Lemma 3.1 shows that under certain conditions, the probability that the OR curvature of between-communities edges being no larger than $\frac{n-n_i-n_j}{n-1}p_{out} + \frac{n_i-n_j-2}{n-1}p_\vee$ goes to 1 as $n$ goes to infinity. Thus, we can conclude Lemma 3.2 that the probability of the maximum difference of two within-community edges' OR curvatures being $\frac{p_{out}(p_{in}^{(i)} - p_{out})}{\frac{n_i-1}{n-n_i}p_{in}^{(i)} + p_{out}}$ goes to 1 as $n$ goes to infinity. Since the largest OR curvature difference of two within-community edges can be bounded, we can prove the difference is small enough to distinguish from the difference of $\kappa(x_{B_i}, y_{B_i})$ and $\kappa(x^*_{B_i}, y^*_{B_j})$.

**Theorem 3.3.** *Combining the conclusion of statement 1 and statement 3 in Lemma 3.1, if $p_\vee < \frac{p_{in}^{(i)} n}{n_i + n_j}$, $p_{in}^{(i)} < \frac{p_{out}(n-n_i)(n+n_i+n_j)}{n^2 - (n_i+n_j)(n-2n_i)}$, and conditions given in the lemma 3.1 are satisfied, almost surely, for any $(x^*_{B_i}, y^*_{B_j}) \in \Delta((x_{B_i}, y_{B_i}))$, we have*

$$\max_{(x'_{B_i}, y'_{B_i}) \in \Delta((x_{B_i}, y_{B_i}))} |\kappa(x_{B_i}, y_{B_i}) - \kappa(x'_{B_i}, y'_{B_i})| < \kappa(x_{B_i}, y_{B_i}) - \kappa(x^*_{B_i}, y^*_{B_j}). \tag{8}$$

The Theorem 3.3 shows the OR curvature of the within-community edge $\kappa(x_{B_i}, y_{B_i})$ is larger than the OR curvature of the between-communities edge $\kappa(x_{B_i}, y_{B_j})$ by the maximum of $|\kappa(x_{B_i}, y_{B_i}) - \kappa(x'_{B_i}, y'_{B_i})|$.

The theorem holds under mild conditions. When $p_{out}$ is small enough, and $p_\vee$ is not too large compared to $p_{out}$, the conditions can be satisfied. When the graph size is large, e.g., thousands of or millions of nodes, the constraint of $p_{out}$ and $p_{in}^{(i)}$ is loosened, and the inequality in Theorem 3.3 can be easily satisfied.

## 3.2 OR CURVATURE GRADIENT AND COMMUNITIES

We define the OR curvature gradient (ORG) as $|\kappa(x^{(j)}, y^{(j)}) - \kappa(x^{(i)}, y^{(i)})|$, where $(x^{(j)}, y^{(j)}) \in \Delta((x^{(i)}, y^{(i)}))$. It is the difference between the OR curvature of two adjacent edges. According to the theoretical results, the within-community ORG $|\kappa(x_{B_i}, y_{B_i}) - \kappa(x'_{B_i}, y'_{B_i})|$ is significantly smaller than the between-communities ORG $|\kappa(x_{B_i}, y_{B_i}) - \kappa(x^*_{B_i}, y^*_{B_j})|$, this motivates us to propose a subsampling algorithm based on ORG, which extracts the community information with theoretical guarantee.

# 4 OR CURVATURE GRADIENT BASED GRAPH SUBSAMPLING

The existing graph subsampling methods usually use degree information and random walk through the graph. However, the subsampled graph tends to leave out minority communities (i.e., a community with a smaller number ) since they tend to sample nodes with high degrees. Due to the lack of taking advantage of the community structure information during the expansion, the available subsampling algorithms usually underestimate $M$. From the previous work, the community structure relates to the geometric phenomenon of the underlying Riemannian Manifold. Thus, we develop a fast and efficient ORG-based subsampling algorithm that is able to traverse the communities of the graph instead of trapping in a single community. We then prove the probability of the proposed ORG-sub taking all communities of the full graph into the subsample converges to one as the subsample size increases. Consequently, the subsampled graph attained by the ORG-sub algorithm can get a larger proportion of the minority communities than other subsampling algorithms and helps estimate $M$ more accurately, as shown in Figure 3. In addition, we theoretically show the estimation of $M$ by the subsampling algorithm converges to true $M$.

## 4.1 SUBSAMPLING ALGORITHM

The ORG carries the community structure information since it differentiates within-community edges and between-communities edges with theoretical guarantees in Theorem 3.3. We can use ORG as a guide for expanding the subgraph in the direction of more communities. First, we randomly sample an edge $(x, y) \in E$ as the starting edge. The probability of obtaining an edge $(x, y)$ at start is $P((x^{(1)}, y^{(1)})) = \frac{1}{|E|}$.

To take advantage of the theoretical properties of the ORG, the ORG-based subsampler expands to the next edge whose OR curvature shows the greatest difference from the OR curvature of the previously taken edge. The edge subsampled in the $i + 1$-th step $(x^{(i+1)}, y^{(i+1)})$ given the edge subsampled in the $i$-th step $(x^{(i)}, y^{(i)})$ can be expressed by:

$$(x^{(i+1)}, y^{(i+1)}) = \underset{(x,y) \in \Delta((x^{(i)}, y^{(i)}))}{\arg\max} |\kappa(x, y) - \kappa(x^{(i)}, y^{(i)})|. \tag{9}$$

Since the difference between the within-community ORG is smaller than the between-community ORG, the proposed subsampler will expand to another community instead of trapping in one community. After the subsampler stops expanding or the subsampling budget is used up, we get subsampled nodes. Then we can get the subgraph induced by the subsampled nodes. Details of the algorithm are in Algorithm 1.

---

**Algorithm 1**

---

**Input:** Graph $G$; Number of nodes to be subsampled $\tilde{n}$; OR curvature of the graph $G$ calculated by parameter $\alpha$; the subsampled node set $S = \emptyset$
**Initialization:** Randomly choose a node $v_0$ as the start of sampling. Here, $t = 0$.
**Cold Start:** Among all neighbors of $v_0$, randomly add $v_1$ to $S$. Here, $t = 1$.
**While:** $|S| < \tilde{n}$
    • Step 1: Given the edge $e_t = (v_{t-1}, v_t)$ selected in the $t$-th step, get the edge curvatures of the $e_t$'s neighboring edges $\Delta(e_t)$.
    • Step 2: Select the edge $e_{t+1}$ in the edge set $\Delta(e_t)$ that has the greatest OGR.
    • Step 3: Add nodes connecting the edge $e_{t+1}$ to the subsampled node set $S$.
    • $t = t + 1$.
**Output:** Subsampled node set $S$ and the induced subgraph $G[S]$

---

## 4.2 ORG-SUB FOR ESTIMATING $M$

Applying the proposed ORG-sub algorithm to the full graph, as shown in Figure 3 (A), we can get a subsample as shown in Figure 3 (B). Compared with the subsample obtained by the degree-based subsampler in Figure 3 (C), the proposed algorithm can subsample more nodes in minority communities. The toy example illustrates that the proposed ORG-sub algorithm preserves the community structure information better. Thus, the subsampled graph can help estimate $M$ of the full graph.

We take multiple subgraphs by applying ORG-sub $r$ times. For each subsampled graph $G[S_i]$ given the sample size $\tilde{n}$ at $i$-th time, we get an estimation of $M$, $\widehat{M}(G[S_i])$, then the mean of $\widehat{M}(G[S_i])$ is calculated as our final estimation of $M$. As for the choice of the estimation algorithm, we use the state-of-the-art network cross-validation method (Li et al., 2020).

Compared with other methods (Wang & Bickel, 2017; Chen & Lei, 2018), network cross-validation is more robust to complicated settings, i.e., the number of communities is large or the community structure is not conspicuous.

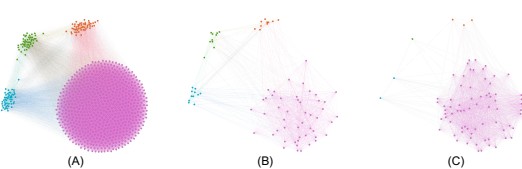

Given the graph $G$ generated by Theorem 3.3, we apply the ORG-sub algorithm 1 to the graph $G$ and take graph subsamples with size $\tilde{n}$ denoted by $G[S]$. Let $v$ denote a node in $G[S]$,

Figure 3: (A): A graph generated by SBM with community size $(650, 50, 50, 50)$, $p_{in} = 0.8$ and $p_{out} = 0.1$. (B): The subsampled graph by the proposed ORG-sub algorithm. The proportion of subsampling is 10%. (C) The subsampled graph by degree-based subsampling algorithm. The proportion of subsampling is 10%.

and $P(v \in B_i)$ denote the probability of the subgraph $G[S]$ carrying nodes in the community $B_i$. Theorem 4.1 shows that the probability of the ORG-sub algorithm expanding to all communities converges to one as the expanding steps increase.

**Theorem 4.1** (Probability of the ORG-sub's Expansion to Communities). *Under the conditions in Theorem 3.3, as $\frac{\tilde{n}}{n} \to 1$, we have $P(v \in B_i \mid v \in G[S]) \to 1$.*

In addition, based on Theorem 4.1, we prove the theoretical advantage of our method over a popular graph sampling method, i.e., random walk sampling (Lovász, 1996). Let $G[S]$ denote a sampled subgraph of $\tilde{n}$ nodes, obtained by the ORG-sub method, and $G_{RW}[S]$ denote a sampled subgraph obtained by the random walk sampling method. Let $v_{RW}$ denote a node in $G_{RW}[S]$. Corollary 4.1.1 shows that $P(v \in B_i \mid v \in G[S])$ is greater than $P(v_{RW} \in B_i \mid v \in G[S])$. This means the ORG-sub is more likely to obtain a subgraph containing nodes from different communities.

**Corollary 4.1.1** (Theoretical advantage over random walk). *Under the assumptions in Theorem 4.1, for any community $B_i$ in graph $G$, we have $P(v_{RW} \in B_i \mid v \in G[S]) < P(v \in B_i \mid v \in G[S])$.*

Theorem 4.1 theoretically guarantees each community in the full graph is subsampled with high probability, thus, the convergence of the estimation of $M$ by the ORG-sub algorithm can be theoretically guaranteed referring to Theorem 3 in (Li et al., 2020). Since cross-validation methods tend not to give $\widehat{M} > M$ in SBMs (Li et al., 2020; Chen & Lei, 2018), we can provide a theoretical guarantee against under-estimation of the proposed ORG-sub-based estimator $\widehat{M}(G[S_i])$.

**Corollary 4.1.2.** *If the subsample with size $\tilde{n}$ satisfies $\frac{\tilde{n}\rho}{\log \tilde{n}} \to \infty$, where $\rho$ is the edge density, as the subsampling times $r \to \infty$, the probability the subsampled graph underestimates the true $M$ is:*

$$Pr(\frac{1}{r}\sum_{i=1}^{r} \widehat{M}(G[S_i]) < M) \to 0, \tag{10}$$

## 4.3 COMPUTATIONAL ANALYSIS OF ORG-SUB

**Computational complexity of Algorithm 1.** We analyze the time complexity of Algorithm 1 step by step. The time complexity of querying the neighboring edges and their corresponding curvatures is of order $\mathcal{O}(n\tilde{d})$ by Nys-sink algorithm (Altschuler et al., 2019), given $\tilde{d}$ is the average degree. The time complexity of the sorting step to attaining the most different curvature from the neighbors is $\mathcal{O}(\tilde{d}\log\tilde{d})$. Thus, the time complexity of taking one step is of order $\mathcal{O}(n\tilde{d})$. To sample $\tilde{n}$ nodes, we need to run about $\tilde{n}$ steps and get the induced subgraph from the sampled node set $S$. Since getting an induced subgraph of size $\tilde{n}$ takes time complexity of order $\mathcal{O}(\tilde{n}\tilde{d})$, the total complexity of the sampling procedure is $\mathcal{O}(n\tilde{n}\tilde{d})$.

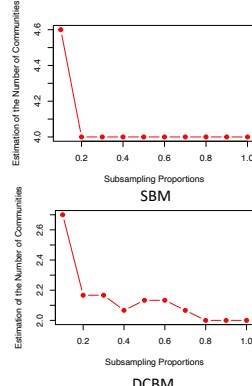

Figure 4: The function of the estimation of $M$ with respect to the subsampling proportion for SBM and DCBM, respectively.

**Choice of the subsample Size.** When implementing our algorithm, we need to specify the size of the subsamples, which determines the performance and efficiency of the algorithm. Empirically, we plot the estimation of $M$ as the function of the subsampling proportion in Figure 4, and we can select the elbow where the estimation of $M$ starts to converge in applications to save computational resources and stabilize the estimation results.

## 5 EXPERIMENT

We evaluate the performance of our algorithm on both synthetic datasets and real-world datasets. We use the metric, MAE$= \frac{1}{r} \sum_{i=1}^{r} \left| M - \widehat{M}(G[S_i]) \right|$ to evaluate the accuracy of subsampling based estimation. We compare the proposed method with (1) Degree-based Node sampler (DBN) (Adamic et al., 2001), (2) Community Structure Expansion Sampler (CSE) (Maiya & Berger-Wolf, 2010) which is a community structure preserved sampler, and six benchmark exploration-based graph subsampling methods, including (3) Metropolis Hasting Random Walk Sampler (MHRW) (Hübler et al., 2008), (4) Forest Fire Sampler (FFS) (Leskovec et al., 2005), (5) Snowball Sampler (Goodman, 1961), (6) Random Walk Sampler (RW) (Gjoka et al., 2010), (7) Multi-dimensional Random Walk Sampler (MDRW) (Ribeiro & Towsley, 2010). As for other methods, we set the hyper-parameters as the default in package *Little Ball of Fur* (Rozemberczki et al., 2020b). We replicate the experiments 30 times under each setting and compare the performance in terms of MAE for all methods. All the experiments are conducted on a machine with a 40-core NVIDIA Tesla V100 GPU (3.00 GHz).

### 5.1 DATASETS

**Synthetic Dataset**    We generate synthetic datasets by the stochastic block models (SBM) and degree-corrected block models (DCBM), which can assign the community distribution to each node. We set the community proportion as $(3/4, 1/10, 1/12, 1/15)$ with 900 nodes in total. The out-in-ratio (the ratio of between-communities edges over within-community edges) controls the ratio of the number of edges between the communities and within a community. A higher ratio represents a noisier graph. The degree-corrected model corrects the node degree by a power-law distribution. Given the probability of an edge within a community ($p_{in} = 0.8$), we vary the probability of an edge between communities ($p_{out}$) ($\{0.06, 0.08, 0.10, 0.12\}$) and subsampling proportions ($\{0.1, 0.12, 0.14, 0.16\}$) from low to high for both block models to observe how our method performs as the setting becomes more challenging compared with others. More details about the generation of the DCBM datasets are presented in section B.1.1 in Appendix.

**Real-world Dataset**    We use five widely-used read-world graph datasets with labeled community structures to validate the performance of our method, including Polbooks, Facebook, Cora, Polblogs, and PubMed (Rossi & Ahmed, 2015; Rozemberczki et al., 2020a; Sen et al., 2008). All these graphs are considered unweighted and undirected. Besides, all the self-loop edges and isolated nodes are removed. Table A.4 in the Appendix summarizes the network statistics of these datasets. As we can see, the number of nodes ranges from 105 to 19,717, and the network density ranges from 0.0001 to 0.04. Thus the networks we considered span a wide range.

### 5.2 RESULTS OF SYNTHETIC DATASET

The subsampling time $r$ for estimating $M$ is set as 3, which is enough to get a stable estimation result (Li et al., 2020). The results of SBM and DCBM datasets are reported in Figure 5. The error bar is the standard deviation of 30 replications. We set the subsampling proportion (prop) as 0.1 when varying the probability of an edge between communities (pout) and set pout=0.06 when varying the prop. Our method can get a more accurate estimation of $M$ than other methods. More details about the table of results, including each combination of pout and prop values, are presented in Table A.1 and A.2 in Appendix. We also compare the computation time for estimating $M$

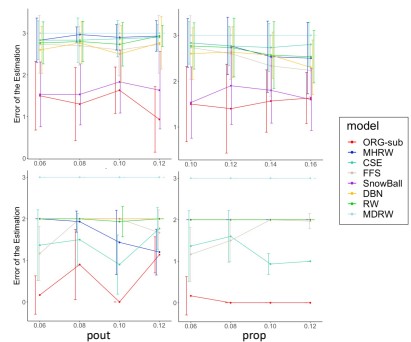

Figure 5: The function of the performance of MAE with respect to pout and prop for DCBM dataset.

by using the whole graph and the subgraph. It turns out that the time for estimating the subgraph together with the time for subsampling is still much shorter than the time for estimation on the whole graph. It is consistent with our conclusion about the complexity of the algorithm. Details about computational time are presented in Section B.1.2 in Appendix.

## 5.3 Results of Real-world Dataset

We compare the performance of the proposed ORG-sub algorithm with subsamples generated by different algorithms by MAE defined above. Table 1 records the average error of the estimation of $M$ over 30 replications. Still, our algorithm outperforms other algorithms in most cases. Let $n_1, \ldots, n_M$ denote the number of nodes in $M$ communities. Here, we calculate the normalized Shannon Entropy of $n_1, \ldots, n_M$ as a metric to measure the imbalance level, i.e., we calculate IM $= -\frac{1}{\log M} \sum_{i=1}^{M} \frac{n_i}{n} \log \frac{n_i}{n}$ and a higher IM value means that the communities are more imbalanced. Table A.4 in the appendix summarizes the IM. As we can see, both PubMed and Polbooks are more imbalanced than Facebook. From Table 1, we observe that the ORG-sub performances of PubMed and Polbooks are better than that of Facebook. This shows that ORG-sub performs better when the community of the graph is more imbalanced. The true $M$ of each dataset is recorded in the first column of Table 1. We tested the performance of different algorithms with a broader range of sampling proportions ranging from 0.5% to 30%. In particular, for small-sized networks Polbooks and Facebook, we consider sampling proportions 10%, 20%, and 30%. For medium-size networks Polblogs and Cora, we consider 5%, 10%, and 20%. For large-size network PubMed, we consider 0.5%, 2%, and 5%. In addition, the computation time on the subgraph (together with the subsampling time) is still much shorter than on the whole graph. Details about computational time are presented in Section B.2.2 in Appendix.

Table 1: Comparison of the performance on the error of the estimation of $M$ for each dataset and subsampling method.

| Dataset | Prop. | ORG-sub | MHRW | CSE | FFS | SnowBall | DBN | RW | MDRW |
|---|---|---|---|---|---|---|---|---|---|
| Polbooks True: 3 | 10% | **0.00** | 1.20 | 0.62 | 2.68 | 0.48 | 0.60 | 0.33 | 0.00 |
| | 20% | **0.00** | 0.19 | 0.52 | 0.30 | 0.70 | 1.76 | 0.37 | 1.60 |
| | 30% | **0.23** | 1.00 | 0.43 | 0.37 | 0.93 | 1.60 | 0.37 | 2.00 |
| Facebook True: 11 | 10% | **5.27** | 6.83 | 7.97 | 6.87 | 8.77 | 7.57 | 5.77 | 10.00 |
| | 20% | 5.67 | 5.13 | 5.93 | 6.93 | 7.20 | **4.23** | 5.50 | 9.90 |
| | 30% | **5.90** | 6.20 | 8.97 | 7.13 | 8.77 | 7.27 | 8.97 | 10.00 |
| Cora True: 7 | 5% | 2.27 | 2.33 | 4.93 | 5.97 | 3.77 | 3.10 | **1.30** | 4.60 |
| | 10% | 3.40 | **1.53** | 4.97 | 3.40 | 3.00 | 3.37 | 3.87 | 5.80 |
| | 20% | **2.53** | 2.97 | 5.00 | 3.90 | 4.80 | 4.80 | 2.73 | 5.47 |
| Polblogs True: 2 | 5% | **0.00** | 1.87 | 0.90 | 2.00 | 0.43 | 1.33 | 1.03 | 0.30 |
| | 10% | **0.00** | 0.40 | 0.33 | 0.20 | 0.03 | 0.03 | 0.07 | 0.87 |
| | 20% | **0.00** | 1.87 | 0.90 | 2.00 | 0.43 | 1.33 | 1.03 | 0.30 |
| PubMed True: 3 | 0.5% | **0.20** | 0.30 | 0.30 | 0.50 | 1.00 | 0.70 | 0.30 | 1.90 |
| | 2% | **0.00** | 0.30 | 0.80 | 0.40 | 0.20 | 0.70 | 1.20 | 1.80 |
| | 5% | **0.30** | 0.55 | 0.55 | 1.40 | 1.65 | 0.65 | 2.20 | 2.75 |

## 6 Conclusion

In this paper, we propose a novel Ollivier-Ricci Curvature Gradient-based graph subsampling (ORG-sub) method that samples a subgraph by maximizing OR curvature gradient. The contribution of the ORG-sub method to the graph subsampling research line is three-fold. First, to the best of our knowledge, we are the first to utilize the graph's internal topological information to subsample a large graph that preserves the number of communities. Second, to the best of our knowledge, we are the first to bridge the gap in the consistency theory regarding subsampling algorithms for the number of community estimations in SBMs. In particular, we theoretically show that ORG-sub effectively traverses different communities and avoids trapping in one community. In addition, we theoretically show the advantage of ORG-sub over a popular sampling method, random walk. Third, we empirically show that our method has superior performance over existing subsampling algorithms in terms of estimating the number of communities. An interesting future direction we plan to investigate is to extend the theory of our method to SBM variants, including degree-correlated SBM, overlapping SBM, and multi-layer SBM. Another future direction is to investigate other community-related statistics that our method has preserved. In fact, we have empirically observed some promising results in preserving the clustering coefficient (CC). Empirical results can be found in Section E in Appendix. In the future, we will investigate the consistency theory of CC.

ACKNOWLEDGMENTS

Thanks to the partial support by NSF awards DMS-1903226, DMS-1925066, DMS-2124493, and NIH grant R01GM1222080. We would like to thank the referees for their valuable comments and suggestions that helped us to improve the quality of this paper. We appreciate the time and effort the referees put into the review process, and we are grateful for their contributions.

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

# Appendix for "Subsampling for Large Graphs Using Ricci Curvature"

The appendix shows the details of the proof, details of experiments, including the parameter settings of generating the synthetic datasets, a description of real-world datasets, and more experiment results on synthetic and real-world datasets.

## A PROOF DETAILS

**Lemma A.1** (Ollivier-Ricci curvature of random graphs generated by stochastic block models). *Consider a graph with $M$ communities generated from a stochastic block model (SBM)* $\mathcal{B}\Big( \big\{p_{in}^{(i)}\big\}_{i=1}^{M}, p_{out}, \{n_i\}_{i=1}^{M} \Big)$*, where $p_{in}^{(i)}$ represents the probability of an edge connecting nodes in the same community $B_i$, $p_{out}$ represents the probability of an edge connecting nodes from different communities, and $\{n_i\}_{i=1}^{M}$ are the sizes of each community in the SBM. We assume that, for any $i = 1, ..., M$, $p_{in}^{(i)} > p_{out}$, and $\frac{n_i}{n}$ converges to $r_i(1 > r_i > 0)$, as $n$ goes to infinity. Let $n_\vee = \max\{n_i\}_{i=1}^{M}$, and $p_\vee = \max\big\{p_{in}^{(i)}\big\}_{i=1}^{M}$. We denote the node $x$ from the community $B_i$ by $x_{B_i}$, the edge $(x_{B_i}, y_{B_i})$ where $x$ is from $B_i$ and $y$ is from $B_j$, and the edge's OR curvature by $\kappa(x_{B_i}, y_{B_i})$. The following statements hold for the OR curvature.*

1. *If $p_{out} \leq \sqrt{\frac{(n_i-1)\ln n}{n \ln n_i}}$ and $p_{in}^{(i)}n_i + p_{out}(n - n_i) > \frac{8}{3}\ln n_i$, for the lower bound of within-community OR curvature, we almost surely have*

$$\kappa(x_{B_i}, y_{B_i}) \geq \frac{(n_i - 2)p_{in}^{(i)^2} + (n - n_i)p_{out}^2}{(n_i - 1)p_{in}^{(i)} + (n - n_i)p_{out}} + \mathcal{O}\Big(\frac{\ln n}{p_{out}^2 n}\Big). \tag{11}$$

2. *If $p_{in}^{(i)^2}n_i + p_{out}^2(n - n_i) > 4\ln n_i$, for the upper bound of within-community OR curvature, we almost surely have*

$$\kappa(x_{B_i}, y_{B_i}) \leq p_{in}^{(i)} + \mathcal{O}\Big(\frac{\ln n_i}{n}\Big). \tag{12}$$

3. *If $p_{out} > \sqrt{(3 \ln n_\vee)\Big/ \Big[\frac{p_{in}^{(i)}}{p_{out}}(n_i + n_j) + n - n_i - n_j\Big]}$, for the upper bound of between-communities OR curvature, we almost surely have*

$$\kappa(x_{B_i}, y_{B_j}) \leq \frac{n - n_i - n_j}{n - 1}p_{out} + \frac{n_i - n_j - 2}{n - 1}p_\vee + \mathcal{O}\Big(\sqrt{\frac{p_\vee \ln n}{p_{out}n}}\Big). \tag{13}$$

We will use Chernoff's inequality to prove the lemmas.

**Lemma A.2.** *(Chernoff's inequality) Let $X_1, ..., X_n$ be the independent random variables with*

$$P(X_i = 1) = p_i, P(X_i = 0) = 1 - p_i \tag{14}$$

*We consider the sum $X = \sum_{i=1}^{n} X_i$, with expectation $E(X) = \sum_{i=1}^{n} p_i$. Then we have*

$$(Lowertail)P(X \leq E(X) - \lambda) \leq e^{-\lambda^2/2\mathrm{E}(X)} \tag{15}$$

$$(Uppertail)P(X \geq E(X) + \lambda) \leq e^{-\lambda^2/(2\mathrm{E}(X)+2\lambda/3)} \tag{16}$$

WLOG, the proof of the theorem considers the situation when $M = 2$. The conclusion can be easily generated to any arbitrary $M$ by setting one block as block $B_1$, and the other blocks as another block $B_2$. Before we prove the theorem, we need a few lemmas.

**Lemma A.3.** *If $p_{in}^{(i)} n_i + p_{out}(n - n_i) > \frac{8}{3} \ln n_i$, then the degree of all nodes in block $B_i$ fall in the range $\left( p_{in}^{(i)}(n_i - 1) + p_{out}(n - n_i)) - \sqrt{4(p_{in}^{(i)} n_i + p_{out}(n - n_i)) \ln n_i}, p_{in}^{(i)}(n_i - 1) + p_{out}(n - n_i) + \sqrt{6(p_{in}^{(i)} n_i + p_{out}(n - n_i)) \ln n_i} \right)$ with probability at least $1 - 2/n_i$.*

*Proof.* Without loss of generality, we just prove the first statement and the second statement can be proved in the same way.

For each vertex $v$ in block $B_i$, it is easy to show that the expected degree of vertex $v$ is $E(d_v^{B_i}) = p_{in}^{(i)}(n_i - 1) + p_{out}(n - n_i)$. Applying Chernoff's inequality with the lower tail $\lambda = \sqrt{4(p_{in}^{(i)} n_i + p_{out}(n - n_i)) \ln n_i}$, we have

$$P\left( d_v^{B_i} - (p_{in}^{(i)}(n_i - 1) + p_{out}(n - n_i)) \leq -\sqrt{4(p_{in}^{(i)} n_i + p_{out}(n - n_i)) \ln n_i} \right) \qquad (17)$$

$$\leq e^{\frac{-4(p_{in}^{(i)} n_i + p_{out}(n - n_i)) \ln n_i}{2(p_{in}^{(i)}(n_i - 1) + p_{out}(n - n_i))}} \leq \frac{1}{n_i^2} \qquad (18)$$

Applying Chernoff's inequality with the upper tail $\lambda = \sqrt{6(p_{in}^{(i)} n_i + p_{out}(n - n_i)) \ln n_i}$, we have

$$P\left( d_v^{B_i} - (p_{in}^{(i)}(n_i - 1) + p_{out}(n - n_i)) \geq \sqrt{6(p_{in}^{(i)} n_i + p_{out}(n - n_i)) \ln n_i} \right) \qquad (19)$$

$$\leq e^{\frac{-6(p_{in}^{(i)} n_i + p_{out}(n - n_i)) \ln n_i}{2(p_{in}^{(i)}(n_i - 1) + p_{out}(n - n_i)) + 2/3 \sqrt{6(p_{in}^{(i)} n_i + p_{out}(n - n_i)) \ln n_i}}} < \frac{1}{n_i^2} \qquad (20)$$

In the last step, we used the assumption $p_{in}^{(i)} n_i + p_{out}(n - n_i) > \frac{8}{3} \ln n_i$. The probability that there is a vertex $v$ in block $B_i$ so that $d_v^{B_i} \notin \left( p_{in}^{(i)}(n_i - 1) + p_{out}(n - n_i)) - \sqrt{4(p_{in}^{(i)} n_i + p_{out}(n - n_i)) \ln n_i}, p_{in}^{(i)}(n_i - 1) + p_{out}(n - n_i) + \sqrt{6(p_{in}^{(i)} n_i + p_{out}(n - n_i)) \ln n_i} \right)$ is at most $n_i \left( \frac{1}{n_i^2} + \frac{1}{n_i^2} \right) = \frac{2}{n_i}$. $\qquad \square$

The co-degree $d_{xy}$ of a pair of vertices $(x, y)$ is the cardinality of the common neighborhood of $x$ and $y$. Roughly speaking, when $p_{in}^{(i)}$ and $p_{out}$ is large, the co-degree follows the binomial distribution; when $p_{in}^{(i)}$ and $p_{out}$ is small, the co-degree follows the Poisson distribution. We can expect that all co-degrees within a block and between blocks are concentrated around a small interval.

**Lemma A.4.** *If $p_{in}^{(i)^2} n_i + p_{out}^2(n - n_i) > 4 \ln n_i$, then with probability at least $1 - 1/n_i$, all co-degrees of a pair of vertices in block $B_i$ fall in the range $\left( p_{in}^{(i)^2}(n_i - 2) + p_{out}^2(n - n_i) - \sqrt{6(p_{in}^{(i)^2} n_i + p_{out}^2(n - n_i)) \ln n_i}, p_{in}^{(i)^2}(n_i - 2) + p_{out}^2(n - n_i) + \sqrt{9(p_{in}^{(i)^2} n_i + p_{out}^2(n - n_i)) \ln n_i} \right)$.*

*If $p_{in}^{(i)} p_{out} n_i + p_{out} p_{in}^{(i)} n_j + p_{out}^2(n - n_i - n_j) > 3 \ln n_\vee$, then with probability at least $1 - 1/n_\vee$, all co-degrees of a pair of vertices in blocks $B_i$ and $B_j$ fall in the range $\left( p_{in}^{(i)} p_{out}(n_i - 1) + p_{out}^2(n - n_j - n_i) + p_{in}^{(i)} p_{out}(n_j - 1) - \sqrt{6(p_{in}^{(i)} p_{out} n_i + p_{out} p_{in}^{(i)} n_j + p_{out}^2(n - n_i - n_j)) \ln n_\vee}, p_{in}^{(i)} p_{out}(n_i - 1) + p_{out}^2(n - n_j - n_i) + p_{in}^{(i)} p_{out}(n_j - 1) + \sqrt{9(p_{in}^{(i)} p_{out} n_i + p_{out} p_{in}^{(i)} n_j + p_{out}^2(n - n_i - n_j)) \ln n_\vee} \right)$.*

*Proof.* For a pair of vertices $x \in B_i$ and $y \in B_i$, the co-degree $d_{x^{B_i} y^{B_i}}$ is the sum of $n - 2$ independent random variables $X_1, ..., X_{n-2}$ with expectation $E(d_{x^{B_i} y^{B_i}}) = p_{in}^{(i)^2}(n_i - 2) + p_{out}^2(n - n_i)$.

Applying Chernoff's inequality with the lower tail $\lambda = \sqrt{6({p_{in}^{(i)}}^2 n_i + p_{out}^2(n - n_i)) \ln n_i}$, we have

$$P\left(d_{x^{B_i}y^{B_i}} - ({p_{in}^{(i)}}^2(n_i - 2) + p_{out}^2(n - n_i)) < -\sqrt{6({p_{in}^{(i)}}^2 n_i + p_{out}^2(n - n_i)) \ln n_i}\right) \quad (21)$$

$$\leq e^{-\frac{6{p_{in}^{(i)}}^2 n_i + p_{out}^2(n-n_i)) \ln n_i}{2({p_{in}^{(i)}}^2(n_i-2) + p_{out}^2(n-n_i))}} \leq \frac{1}{n_i^3}. \quad (22)$$

If ${p_{in}^{(i)}}^2 n_i + p_{out}^2(n - n_i) > 4 \ln n_i$, we apply Chernoff's inequality with the upper tail $\lambda = \sqrt{9({p_{in}^{(i)}}^2 n_i + p_{out}^2(n - n_i)) \ln n_i}$,

$$P\left(d_{x^{B_i}y^{B_i}} - ({p_{in}^{(i)}}^2(n_i - 2) + p_{out}^2(n - n_i)) > \sqrt{9({p_{in}^{(i)}}^2 n_i + p_{out}^2(n - n_i)) \ln n_i}\right) \quad (23)$$

$$\leq e^{-\frac{9({p_{in}^{(i)}}^2 n_i + p_{out}^2(n-n_i)) \ln n_i}{2({p_{in}^{(i)}}^2(n_i-2) + p_{out}^2(n-n_i)) + 2/3\sqrt{9({p_{in}^{(i)}}^2 n_i + p_{out}^2(n-n_i)) \ln n_i}}} < \frac{1}{n_i^3}. \quad (24)$$

Now the number of pairs is at most $n_i^2/2$. the sum of the probabilities of small events is at most $\frac{n_i^2}{2}\left(\frac{1}{n_i^3} + \frac{1}{n_i^3}\right) = \frac{1}{n_i}$.

For a pair of vertices $x \in B_i$ and $y \in B_j$, the co-degree $d_{x^{B_i}y^{B_j}}$ is the sum of $n - 2$ independent random variables $X_1, ..., X_{n-2}$ with expectation $E(d_{x^{B_i}y^{B_j}}) = p_{in}^{(i)}p_{out}(n_i - 1) + p_{out}^2(n - n_j - n_i) + p_{in}^{(i)}p_{out}(n_j - 1)$. Applying Chernoff's inequality with the lower tail $\lambda = \sqrt{6(p_{in}^{(i)}p_{out}n_i + p_{out}p_{in}^{(i)}n_j + p_{out}^2(n - n_i - n_j)) \ln n_\vee}$, we have

$$P\left(d_{x^{B_1}y^{B_2}} - (p_{in}^{(i)}p_{out}(n_i - 1) + p_{out}^2(n - n_j - n_i) + p_{in}^{(i)}p_{out}(n_j - 1)) \right. \quad (25)$$

$$\left. < -\sqrt{6(p_{in}^{(i)}p_{out}n_i + p_{out}p_{in}^{(i)}n_j + p_{out}^2(n - n_i - n_j)) \ln n_\vee}\right) \quad (26)$$

$$\leq e^{-\frac{(6 \ln n_\vee)(p_{in}^{(i)}p_{out}(n_i-1) + p_{out}^2(n-n_j-n_i) + p_{in}^{(i)}p_{out}(n_j-1))}{2(p_{in}^{(i)}p_{out}(n_i-1) + p_{out}^2(n-n_j-n_i) + p_{in}^{(i)}p_{out}(n_j-1))}} \leq \frac{1}{n_\vee^3}. \quad (27)$$

If $p_{in}^{(i)}p_{out}n_i + p_{out}p_{in}^{(i)}n_j + p_{out}^2(n - n_i - n_j) > 4 \ln n_\vee$, we apply Chernoff's inequality with the upper tail $\lambda = \sqrt{9(p_{in}^{(i)}p_{out}n_i + p_{out}p_{in}^{(i)}n_j + p_{out}^2(n - n_i - n_j)) \ln n_\vee}$,

$$P\left(d_{x^{B_i}y^{B_j}} - (p_{in}^{(i)}p_{out}(n_i - 1) + p_{out}^2(n - n_j - n_i) + p_{in}^{(i)}p_{out}(n_j - 1)) \right. \quad (28)$$

$$\left. > \sqrt{9(p_{in}^{(i)}p_{out}n_i + p_{out}p_{in}^{(i)}n_j + p_{out}^2(n - n_i - n_j)) \ln n_\vee}\right) \quad (29)$$

$$\leq e^{-\frac{9(p_{in}^{(i)}p_{out}n_i + p_{out}p_{in}^{(i)}n_j + p_{out}^2(n-n_i-n_j)) \ln n_\vee}{2(p_{in}^{(i)}p_{out}n_i + p_{out}p_{in}^{(i)}n_j + p_{out}^2(n-n_i-n_j)) + 2/3\sqrt{9(p_{in}^{(i)}p_{out}n_i + p_{out}p_{in}^{(i)}n_j + p_{out}^2(n-n_i-n_j)) \ln n_\vee}}} \quad (30)$$

$$< \frac{1}{n_\vee^3}. \quad (31)$$

The number of pairs is at most $n_\vee^2$. the sum of the probabilities of small events is at most $(n_\vee^2)\left(\frac{1}{n_\vee^3}\right) = \frac{2}{n_\vee}$. $\qquad \square$

Referring to the lemma mentioned in Lin et al. (2011), we can construct the lower and upper bound of the Ricci curvature of graphs generated by the stochastic block model.

**Lemma A.5.** *Suppose that $\phi : \Gamma(x) \backslash N(y) \to \Gamma(y) \backslash N(x)$ is an injective mapping. Then we have*

$$\kappa(x, y) \geq 1 - \frac{1}{d_y} \sum_{u \in \Gamma(x) \backslash N(y)} d(u, \phi(u)) + \frac{1}{d_x} - \frac{3(d_y - d_x)}{d_y}.$$

$$\kappa(x,y) = \lim_{\alpha \to 1} \frac{1 - W(m_x^\alpha, m_y^\alpha)}{1 - \alpha} \leq \frac{d_{xy} + 1}{d_x} + \frac{1}{d_y}.$$

**Lemma A.6.** *For the lower bound, we will construct a matching $M$ from $\Gamma(x) \backslash N(y)$ to $\Gamma(y) \backslash N(x)$ as follows. Let $U_0 = \Gamma(x) \backslash N(y)$ and $V_0 = \Gamma(y) \backslash N(x)$. Pick up a vertex $u_1 \in U_0$. Reveal the neighborhood of $u_1$ in $V_0$. Pick a vertex in the neighborhood, and denote it by $v_1$. Let $U_1 = U_0 \backslash \{u_1\}$ and $V_1 = V_0 \backslash \{v_1\}$ and continue this process. The process ends when $\Gamma(u_{i+1}) \cap V_i = \emptyset$. The probability that the maximum matching between $U_0$ and $V_0$ is at most $k$ is less than*

$$\sum_{i=1}^{k} (1 - p_{out})^{|V_0| - i} < \frac{1}{p}(1 - p_{out})^{|V_0| - k} \leq \sqrt{\frac{n}{\ln n}} e^{-p_{out}(|V_0| - k)} \leq n e^{-p_{out}(|V_0| - k)}.$$

*Choose $k = \lfloor |V_0| - (3 \ln n)/p \rfloor$. With probability at least $1 - 1/n^2$, there is a matching $M$ of size $k$ between $\Gamma(x) \backslash N(y)$ and $\Gamma(y) \backslash N(x)$.*

Now we extend the matching $M$ to an injective mapping $\phi : \Gamma(x) \backslash N(y) \to \Gamma(y) \backslash N(x)$ arbitrarily. Applying Lemmas A.6, with probability at least $1 - 4/n$, we have

$$\begin{aligned}
\kappa(x,y) &\geq 1 - \frac{1}{d_y} \sum_{u \in \Gamma(x) \backslash N(y)} d(u, \phi(u)) + \frac{1}{d_x} - \frac{3(d_y - d_x)}{d_y} \\
&\geq 1 - \frac{1}{d_y}(k + 3(|V_0| - k)) + \frac{1}{d_x} - \frac{3(d_y - d_x)}{d_y} \\
&\geq \frac{d_{xy}}{d_y} - \frac{2(3 \ln n / p_{out})}{d_y} - \frac{3(d_y - d_x)}{d_y}.
\end{aligned}$$

*Proof.* Given the range of the degrees and co-degrees, lower bound and upper bound of Ricci curvatures with respect to $d_x$ and $d_{xy}$, we can prove the remark 3 of Lemma A.1 as follows.

For vertex $x \in B_i, y \in B_i$, given $p_{in}^{(i)} n_i + p_{out}(n - n_i) > \frac{8}{3} \ln n_i$, we have

$$\begin{aligned}
\kappa(x_{B_i}, y_{B_i}) &\leq \frac{(n_i - 2)p_{in}^{(i)^2} + (n - n_i)p_{out}^2 + \sqrt{9 \ln n_1 (n_1 p_{in}^{(i)^2} + n_2 p_{out}^2)} + 2}{(n_i - 1)p_{in}^{(i)} + (n - n_i)p_{out} - \sqrt{4(n_i p_{in}^{(i)} + (n - n_i)p_{out}) \ln n_i}} \\
&= p_{in}^{(i)} + \mathcal{O}(\frac{\ln n_i}{n}).
\end{aligned}$$

For vertex $x \in B_i, y \in B_i$, given $p_{in}^{(i)^2} n_i + p_{out}^2(n - n_i) > 4 \ln n_i$, we have

$$\kappa(x_{B_i}, y_{B_i}) \geq \frac{(n_i - 2)p_{in}^{(i)^2} + (n - n_i)p_{out}^2 - \sqrt{6 \ln n_i (n_i p_{in}^{(i)^2} + (n - n_i)p_{out}^2)}}{(n_i - 1)p_{in}^{(i)} + (n - n_i)p_{out} + \sqrt{6 \ln n_i (n_i p_{in}^{(i)} + (n - n_i)p_{out})}}$$

$$- \frac{6 \ln n / p_{out}}{(n_i - 1)p_{in}^{(i)} + n_2 p_{out} + \sqrt{4 \ln n_i (n_i p_{in}^{(i)} + (n - n_i)p_{out})}}$$

$$- \frac{6\sqrt{6 \ln n_i (n_i p_{in}^{(i)} + (n - n_i)p_{out})}}{(n_i - 1)p_{in}^{(i)} + (n - n_i)p_{out} + \sqrt{4 \ln n_i (n_i p_{in}^{(i)} + (n - n_i)p_{out})}}$$

$$\geq \frac{(n_i - 2)p_{in}^{(i)^2} + (n - n_i)p_{out}^2}{(n_i - 1)p_{in}^{(i)} + (n - n_i)p_{out}} - \mathcal{O}\left(\frac{\ln n}{p_{out}^2 n}\right) - \mathcal{O}\left(\sqrt{\frac{p_{in}^{(i)} \ln n_i}{p_{out}^2 n}}\right)$$

$$= \frac{(n_i - 2)p_{in}^{(i)^2} + (n - n_i)p_{out}^2}{(n_i - 1)p_{in}^{(i)} + (n - n_i)p_{out}} - \mathcal{O}\left(\frac{\ln n}{p_{out}^2 n}\right) - \mathcal{O}\left(\sqrt{\frac{p_{in}^{(i)} \ln n_i}{p_{out}^2 n}}\right).$$

If $p_{out}^2 p_{in}^{(i)} \leq \frac{(\ln n)^2}{n \ln n_i}$, we have $\kappa(x_{B_i}, y_{B_i}) = \frac{(n_i-2)p_{in}^{(i)^2} + (n-n_i)p_{out}^2}{(n_i-1)p_{in}^{(i)} + (n-n_i)p_{out}} - \mathcal{O}\left(\frac{\ln n}{p_{out}^2 n}\right)$.

For vertex $x \in B_i, y \in B_j$, let $n_\vee = \max\{n_i\}_{i=1}^M$, and $p_\vee = \max\left\{p_{in}^{(i)}\right\}_{i=1}^M$. If $p_{out} > \sqrt{(3 \ln n_\vee)\Big/ \left[\frac{p_{in}^{(i)}}{p_{out}}(n_i + n_j) + n - n_i - n_j\right]}$, we have

$$\kappa(x_{B_i}, y_{B_j}) \leq \frac{3\sqrt{\ln n_\vee [(n_i - 1)p_{in}^{(i)}p_{out} + (n - n_i - n_j)p_{out}^2 + (n_j - 1)p_{in}^{(j)}p_{out}]}}{(n_i - 1)p_{in}^{(i)} + n_j p_{out} - 2\sqrt{\ln n_\vee (n_i p_{in}^{(i)} + n_j p_{out})}}$$

$$+ \frac{1 + (n_i - 1)p_{in}^{(i)}p_{out} + (n - n_i - n_j)p_{out}^2 + (n_j - 1)p_{in}^{(j)}p_{out}}{(n_i - 1)p_{in}^{(i)} + n_j p_{out} - 2\sqrt{\ln n_\vee (n_i p_{in}^{(i)} + n_j p_{out})}}$$

$$+ \frac{1}{(n_j - 1)p_{in}^{(j)} + n_i p_{out} - 2\sqrt{\ln n_\vee (n_j p_{in}^{(j)} + n_i p_{out})}}$$

$$\leq \frac{3\sqrt{\ln n_\vee [(n_i - 1)p_{in}^{(i)}p_{out} + (n - n_i - n_j)p_{out}^2 + (n_j - 1)p_{in}^{(j)}p_{out}]} + 2}{(n - 1)p_{out} - 2\sqrt{np_\vee \ln n}}$$

$$+ \frac{(n_i - 1)p_{in}^{(i)}p_{out} + (n - n_i - n_j)p_{out}^2 + (n_j - 1)p_{in}^{(j)}p_{out}}{(n - 1)p_{out} - 2\sqrt{np_\vee \ln n}}$$

$$\leq \mathcal{O}\left(\sqrt{\frac{n_\vee (p_{in}^{(i)} + p_{in}^{(j)} + p_{out}) \ln n_\vee}{(n - 1)^2 p_{out}}}\right) + \frac{n_i + n_j - 2}{n - 1}p_\vee + \frac{n - n_i - n_j}{n - 1}p_{out}$$

$\square$

**Lemma A.7.** *Combining the conclusion of statement 1 and statement 2 in Lemma A.1, almost surely, for any $(x'_{B_i}, y'_{B_j}) \in \Delta((x_{B_i}, y_{B_i}))$, we have*

$$|\kappa(x_{B_i}, y_{B_i}) - \kappa(x'_{B_i}, y'_{B_i})| \leq \frac{p_{out}(p_{in}^{(i)} - p_{out})}{\frac{n_i - 1}{n - n_i}p_{in}^{(i)} + p_{out}} + \mathcal{O}\left(\frac{1}{p_{out}n}\right). \tag{32}$$

*Proof.* We can prove the Lemma A.7 by the lower bound and upper bound of the within-community edges' OR curvature from Lemma A.1. $\square$

**Theorem A.8.** *Combining the conclusion of statement 1 and statement 3 in Lemma A.1, if $p_\vee <$ $\frac{p_{in}^{(i)}n}{n_i+n_j}$, $p_{in}^{(i)} < \frac{p_{out}(n-n_i)(n+n_i+n_j)}{n^2-(n_i+n_j)(n-2n_i)}$ and conditions given in Lemma A.1, almost surely, for any $(x^*_{B_i}, y^*_{B_j}) \in \Delta((x_{B_i}, y_{B_i}))$, we have*

$$\max_{(x'_{B_i}, y'_{B_i}) \in \Delta((x_{B_i}, y_{B_i}))} |\kappa(x_{B_i}, y_{B_i}) - \kappa(x'_{B_i}, y'_{B_i})| < \kappa(x_{B_i}, y_{B_i}) - \kappa(x^*_{B_i}, y^*_{B_j}). \tag{33}$$

*Proof.* We first get the lower bound of the OR curvature difference of within community edge $\kappa(x_{B_i}, y_{B_i})$ and between community edge $\kappa(x^*_{B_i}, y^*_{B_j})$, which can be obtained by the lower bound of $\kappa(x_{B_i}, y_{B_i})$ and upper bound of $\kappa(x^*_{B_i}, y^*_{B_j})$ from the Lemma A.1.

Given the upper bound of the OR curvature difference of edges in the same community, we can prove under certain conditions, the upper bound is smaller than the lower bound we obtained in the last step. Denote $r_i = n_i/n$ and $r_j = n_j/n$.

$$\kappa(x_{B_i}, y_{B_i}) - \kappa(x^*_{B_i}, y^*_{B_j}) - |\kappa(x_{B_i}, y_{B_i}) - \kappa(x'_{B_i}, y'_{B_i})| \tag{34}$$

$$= \frac{(n_i-2)p_{in}^{(i)2} + (n-n_i)p_{out}^2}{(n_i-1)p_{in}^{(i)} + (n-n_i)p_{out}} - \left(\frac{n-n_i-n_j}{n-1}p_{out} + \frac{n_i-n_j-2}{n-1}p_\vee\right) - \frac{p_{out}(p_{in}^{(i)} - p_{out})}{\frac{n_i-1}{n-n_i}p_{in}^{(i)} + p_{out}} \tag{35}$$

$$= \frac{r_i p_{in}^{(i)2} + (1-r_i)p_{out}^2}{r_i p_{in}^{(i)} + (1-r_i)p_{out}} - p_{out} - (r_i+r_j)(p_\vee - p_{out}) - \frac{(1-r_i)p_{out}(p_{in}^{(i)} - p_{out})}{r_i p_{in}^{(i)} + (1-r_i)p_{out}} + \mathcal{O}(\frac{1}{n}) \tag{36}$$

$$= \frac{r_i p_{in}^{(i)2} + 2(1-r_i)p_{out}^2 - (1-r_i)p_{out}p_{in}^{(i)} - r_i p_{in}^{(i)}p_{out} - (1-r_i)p_{out}^2}{r_i p_{in}^{(i)} + (1-r_i)p_{out}} \tag{37}$$

$$- \frac{(r_i+r_j)r_i(p_\vee - p_{out})p_{in}^{(i)} - (r_i+r_j)(1-r_i)p_{out}(p - p_{out})}{r_i p_{in}^{(i)} + (1-r_i)p_{out}} + \mathcal{O}(\frac{1}{n}) \tag{38}$$

$$> \frac{p_{in}^{(i)}r_i(p_{in}^{(i)} - p_\vee(r_i+r_j)) + p_{out}(p_{out}(1-r_i)(1+r_i+r_j) - p_\vee(1-(r_i+r_j)(1-2r_i)))}{r_i p_{in}^{(i)} + (1-r_i)p_{out}}$$

$$\tag{39}$$

$$+ \mathcal{O}(\frac{1}{n}) > 0 \tag{40}$$

$$\square$$

**Theorem A.9** (Probability of the ORG-sub's Expansion to Communities). *Given the graph $G$ generated by Theorem A.8, we apply the ORG-sub Algorithm 1 to the graph $G$ and take graph subsamples with size $\tilde{n}$ denoted by $G[S]$. Under the conditions in Theorem A.8, and if $\tilde{n}/n \to 1$, we can prove that, the probability of the final subgraph $G[S]$ taken by ORG-sub containing nodes in the community $B_i$ is:*

$$P(v \in B_i \mid v \in G[S]) \to 1 \tag{41}$$

*Proof.* Without loss of generality, we show the theorem holds for a two-block stochastic block model (SBM).

We assume the graph is generated from a two-block stochastic block model (SBM), where the probability of an edge within a block is $p_{in}$, and the probability of an edge between the blocks is $p_{out}$. We denote the subsampled graph from the original graph $G$ by $G[S]$, where $S$ is a subset of the full vertices set $V$ of graph $G$, and $G[S]$ is the subgraph induced by the subset $S$. Denote the probability matrix of generating the adjacency matrix $A$ by $\mathbf{P}$:

$$\mathbf{P}^* = \begin{bmatrix} p_{in}\mathbf{1}_{n_1}\mathbf{1}_{n_1}^T & p_{out}\mathbf{1}_{n_1}\mathbf{1}_{n_2}^T \\ p_{out}\mathbf{1}_{n_2}\mathbf{1}_{n_1}^T & p_{in}\mathbf{1}_{n_2}\mathbf{1}_{n_2}^T \end{bmatrix},$$

$$\mathbf{P} = \mathbf{P}^* - diag\{\mathbf{P}^*\}, \tag{42}$$

where the size of the first community is $n_1$, the size of the second community is $n_2$, and the number of nodes of the graph is $n = n_1 + n_2$. The node set belonging to the community 1 is $B_1 = \{v_1, ..., v_{n_1}\}$, and the node set belonging to the community 2 is $B_2 = \{v_{n_1+1,...,n}\}$. Denote each entry of the probability matrix $P$ by $p_{ij}$, then the observed $A$ is generated from:

$$A_{ij} \overset{\text{i.i.d.}}{\sim} \text{Bernouli}(p_{ij}), \tag{43}$$

Define the block of the matrix $A$ as $A^{o_1 o_2} = \{e_{ij} = (v_i, v_j) | v_i \in B_{o_1}, v_j \in B_{o_2}\}$, where $o_1, o_2 \in \{1, 2\}$. We denote the edge subsampled at the $l$-th step by $e_{ij}^{(l)}$, and the neighboring edge set of edge $e_{ij}$ by $\Delta(e_{ij})$.

For each step, the probability of subsampling an edge in a block is a constant. After subsampling one edge in each step, the nodes connected by the subsampled edges are subsampled into node set $S$. We denote the probability of adding node $\tilde{v}$ at the step $l$ belonging to community $B_o$, where $o \in \{1, 2\}$, by $P(\tilde{v}^{(l)} \in B_o)$. After $\tilde{n}$ steps, the probability of the obtained subsample $G[S]$ containing node $\tilde{v}$ from the node set $B_1$ is:

$$
\begin{aligned}
P(\tilde{v} \in B_1 \mid v \in G[S]) =& 1 - P(\{\tilde{v}^{(1)}, \tilde{v}^{(2)}, \cdots, \tilde{v}^{(\tilde{n})}\} \notin B_1) \\
=& 1 - P(e_{ij}^{(\tilde{n}-2)} \in A^{22} | e_{ij}^{(\tilde{n}-3)} \in A^{22}) \cdot P(e_{ij}^{(\tilde{n}-3)} \in A^{22} | e_{ij}^{(\tilde{n}-4)} \in A^{22}) \cdots \\
& P(e_{ij}^{(1)} \in A^{22} | P(e_{ij}^{(0)} \in A^{22}) \cdot P(e_{ij}^{(0)} \in A^{22}) \\
=& 1 - P(\Delta(e_{ij}^{(\tilde{n}-3)}) \subset A^{22} | \Delta(e_{ij}^{(\tilde{n}-4)}) \subset A^{22}) \cdots \\
& P(\Delta(e_{ij}^{(0)}) \subset A^{22} | e_{ij}^{(0)} \in A^{22}) \cdot P(e_{ij}^{(0)} \in A^{22}) \\
=& 1 - \left[(1 - p_{out})^{n_1}(1 - (1 - p_{in})^{n_2 - 2})\right]^{\tilde{n}-2} \cdot P(e_{ij}^{(0)} \in A^{22}).
\end{aligned}
$$

Analogously, the probability of subsampling node $\tilde{v}$ belonging to the node set $B_2$ after $n$ steps is:

$$
\begin{aligned}
P(\tilde{v} \in B_2 \mid v \in G[S]) =& 1 - P(\{\tilde{v}^{(1)}, \tilde{v}^{(2)}, \cdots, \tilde{v}^{(\tilde{n})}\} \notin B_2) \\
=& 1 - P(e_{ij}^{(\tilde{n}-1)} \in A^{11} | e_{ij}^{(\tilde{n}-2)} \in A^{11}) \cdot P(e_{ij}^{(\tilde{n}-2)} \in A^{11} | e_{ij}^{(\tilde{n}-3)} \in A^{11}) \cdots \\
& P(e_{ij}^{(1)} \in A^{11} | P(e_{ij}^{(0)} \in A^{11}) \cdot P(e_{ij}^{(0)} \in A^{11}) \\
=& 1 - P(\Delta(e_{ij}^{(\tilde{n}-3)}) \subset A^{11} | \Delta(e_{ij}^{(\tilde{n}-4)}) \subset A^{11}) \cdots \\
& P(\Delta(e_{ij}^{(0)}) \subset A^{11} | e_{ij}^{(0)} \in A^{11}) \cdot P(e_{ij}^{(0)} \in A^{11}) \\
=& 1 - \left[(1 - p_{out})^{n_2}(1 - (1 - p_{in})^{n_1 - 2})\right]^{\tilde{n}-2} \cdot P(e_{ij}^{(0)} \in A^{11}).
\end{aligned}
$$

Since $(1 - p_{out})^{n_1}(1 - (1 - p_{in})^{n_2-2})$ $((1 - p_{out})^{n_2}(1 - (1 - p_{in})^{n_1-2}))$ can be small when $n_1$ $(n_2)$ and $\tilde{n}$ is large, the proposed ORG-sub subsampler can traverse to new communities with high probability. When $p_{out}$ is small, that is, the edges density between communities is relatively small, the proposed ORG-sub needs more steps to expand to another community. As the traveling steps of the proposed subsampler ORG-sub increase, we can prove that the probability of the proposed ORG-sub algorithm taking all communities into the final subgraph converges to one under the two-stochastic-block model.

In general cases, the graph is generated by SBM with multiple communities, we can always think of the community that the subsampler has never been to as one block, and other communities as the other block. Given the conclusion under the two-stochastic-block model, the ORG-sub can gradually expand the subsampled graph to the block we have never been to. Thus, we can prove the ORG-sub can expand to any community under a multiple-stochastic-block model. □

**Corollary A.9.1** (Theoretical advantage over random walk). *Under the assumptions in Theorem 4.1, for any community $B_i$ in graph $G$, we have*

$$P(v_{RW} \in B_i \mid v \in G[S]) < P(v \in B_i \mid v \in G[S]). \tag{44}$$

*Proof.* For the proposed method, Theorem A.9 has proved that after walking $\tilde{n}$ steps, the probability of the obtained subsample $G[S]$ by the proposed ORG-sub containing vertexes $v$ from community

$B_i$ is:

$$P(v \in B_i \mid v \in G[S]) = 1 - P(e_{ij}^{(\tilde{n}-2)} \in A^{\beta\beta}|e_{ij}^{(\tilde{n}-3)} \in A^{\beta\beta}) \cdot P(e_{ij}^{(\tilde{n}-3)} \in A^{\beta\beta}|e_{ij}^{(\tilde{n}-4)} \in A^{\beta\beta}) \cdots$$
$$P(e_{ij}^{(1)} \in A^{\beta\beta}|P(e_{ij}^{(0)} \in A^{\beta\beta}) \cdot P(e_{ij}^{(0)} \in A^{\beta\beta})$$
$$= 1 - P(\Delta(e_{ij}^{(\tilde{n}-3)}) \subset A^{\beta\beta}|\Delta(e_{ij}^{(\tilde{n}-4)}) \subset A^{\beta\beta}) \cdots$$
$$P(\Delta(e_{ij}^{(0)}) \subset A^{\beta\beta}|e_{ij}^{(0)} \in A^{\beta\beta}) \cdot P(e_{ij}^{(0)} \in A^{\beta\beta}).$$

After $\tilde{n}$ steps, the probability of the obtained subsample $G_{RW}[S]$ by the random walk subsampler containing vertexes in community $B_i$ is

$$P(v_{RW} \in B_i \mid v \in G[S]) = 1 - P(e_{ij}^{(\tilde{n}-2)} \in A^{\beta\beta}|e_{ij}^{(\tilde{n}-3)} \in A^{\beta\beta}) \cdot P(e_{ij}^{(\tilde{n}-3)} \in A^{\beta\beta}|e_{ij}^{(\tilde{n}-4)} \in A^{\beta\beta}) \cdots$$
$$P(e_{ij}^{(1)} \in A^{\beta\beta}|P(e_{ij}^{(0)} \in A^{22}) \cdot P(e_{ij}^{(0)} \in A^{\beta\beta})$$
$$= 1 - \Big[ P(\Delta(e_{ij}^{(\tilde{n}-3)}) \setminus A^{\beta\beta} \neq \varnothing|\Delta(e_{ij}^{(\tilde{n}-4)}) \subset A^{\beta\beta}) +$$
$$P(\Delta(e_{ij}^{(\tilde{n}-3)}) \subset A^{\beta\beta}|\Delta(e_{ij}^{(\tilde{n}-4)}) \subset A^{\beta\beta}) \Big] \cdots$$
$$\Big[ P(\Delta(e_{ij}^{(0)}) \setminus A^{\beta\beta} \neq \varnothing|\Delta(e_{ij}^{(0)}) \subset A^{\beta\beta}) +$$
$$P(\Delta(e_{ij}^{(0)}) \subset A^{\beta\beta}|e_{ij}^{(0)} \in A^{\beta\beta}) \Big] \cdot P(e_{ij}^{(0)} \in A^{\beta\beta}).$$

$\square$

**Corollary A.9.2.** *If the subsample with size $\tilde{n}$ satisfies $\frac{\tilde{n}\rho}{\log \tilde{n}} \to \infty$, where $\rho$ is the edge density, as the subsampling times $r \to \infty$, the probability the subsampled graph underestimates the true $M$ is:*

$$Pr(\frac{1}{r}\sum_{i=1}^{r}\widehat{M}(G[S_i]) < M) \to 0, \tag{45}$$

*Proof.* From Theorem A.9, we know that, if we take enough steps for subsampling, the true number of communities $M(G[S])$ of subgraph $G[S]$ is the same as the true number of communities $M$ of the original graph $G$. Theorem 3 of Li et al. (2020) proves the one-sided consistency of estimating the number of communities of the stochastic block model under the following assumptions. First, the expected node degree $\lambda_n = n\rho$ has the order $\frac{\lambda_n}{\log n}$, where $n$ is the number of nodes of the graph $G$. Second, there exists a constant $\gamma$ such that $\min_k n_k \geq \gamma N$, where $n_k$ is the size of the $k$-th community. Since our subsampled graph satisfies $\frac{\tilde{n}\rho}{\log \tilde{n}} \to \infty$, we can easily verify that the subsampled subgraph satisfies the assumptions in Theorem 3 of Li et al. (2020). The estimate of the number of communities $\widehat{M}(G[S_i])$ corresponding to $S_i$, the $i$-th subsample, satisfies:

$$Pr(\widehat{M}(G[S_i]) < M(G[S_i])) \to 0, \ Pr(\widehat{M}(G[S_i]) < M) \to 0, \tag{46}$$

As the replications of the subsamples increase, it's trivial to show:

$$Pr(\frac{1}{r}\sum_{i=1}^{r}\widehat{M}(G[S_i]) < M) \to 0. \tag{47}$$

$\square$

# B    DETAILS OF EXPERIMENTS

We evaluate the performance of our algorithm on both synthetic datasets and real-world datasets using the metrics presented in the main context. We also compare the proposed method with seven benchmark exploration-based graph subsampling methods, including Metropolis Hasting Random Walk Sampler (MHRW), Forest Fire Sampler (FFS), Snowball Sampler, Community Structure Expansion Sampler (CSE), degree-based node sampler (DBN), Random Walk Sampler (RW) and Multi-dimensional Random Walk Sampler (MDRW). We set the hyper-parameters of the seven

methods as the default in the package *Little Ball of Fur*. The rejection constraint of the MHRW method is set as 1; the burning probability of FFS is 0.4; the bound on the degree of Snowball Sampler is set as 50. The hyper-parameter used to calculate the Ricci curvature of graphs is set as $\alpha = 0.5$. The times of subsampling $r$ in Algorithm 2 for estimation of the number of communities is set as 3, which is enough to get a stable estimation result. All the experiments are conducted on a workstation with a 40-core NVIDIA Tesla V100 GPU (3.00 GHz).

## B.1 EXPERIMENTS ON SYNTHETIC DATASETS

### B.1.1 PARAMETERS SETTINGS

We use stochastic block models (SBM), and degree corrected block models (DCBM), which can assign the community distribution to each node. To create graphs with unbalanced communities, we set the community proportion as $(3/4, 1/10, 1/12, 1/15)$ with 900 nodes. The out-in-ratio controls the ratio of an edge's probability of between-communities and within-community. A higher out-in-ratio represents a noisier graph, i.e., harder to distinguish the community from the graph. We set the probability of an edge within a block as 0.8 and vary the probability of an edge between blocks (pout) for both SBM and DCBM ($\{0.06, 0.08, 0.10, 0.12\}$). The higher the pout value (higher the out-in-ratio) is, the noisier the subsampled graph is. We also change the subsampling proportion ($\{0.1, 0.12, 0.14, 0.16\}$) from low to high for both block models to observe how our method performs compared with others as the settings change. The degree corrected model corrects the node degree using a power-law distribution. More parameters need to be set before generating the graphs following DCBM. We set the average node degree as $40$, with is consistent with the setting in Li et al. (2020), in which the assumptions in our algorithm hold. The node degree follows a power-law distribution with the lower bound as 1 and the scaling parameter as 5.

### B.1.2 ADDITIONAL RESULTS OF SYNTHETIC DATASETS

Table A.1: **DCBM**: Error of the estimation of the number of communities for different subsampling methods under different settings.

| Pout | Prop | ORG-sub | MHRW | CSE | FFS | SnowBall | DBN | RW | MDRW |
|------|------|---------|------|------|------|----------|------|------|------|
| | 0.10 | **1.50** | 2.83 | 2.83 | 2.73 | 1.53 | 2.60 | 2.77 | 3.00 |
| 0.06 | 0.12 | **1.40** | 2.77 | 2.77 | 2.60 | 1.90 | 2.63 | 2.73 | 3.00 |
| | 0.14 | **1.57** | 2.53 | 2.73 | 2.33 | 1.80 | 2.57 | 2.57 | 3.00 |
| | 0.16 | **1.63** | 2.50 | 2.80 | 2.23 | 1.80 | 2.30 | 2.53 | 3.00 |
| | 0.10 | **1.30** | 2.97 | 2.83 | 2.70 | 1.53 | 2.77 | 2.80 | 3.00 |
| 0.08 | 0.12 | **1.57** | 2.77 | 2.83 | 2.77 | 1.80 | 2.53 | 2.53 | 3.00 |
| | 0.14 | **1.57** | 2.90 | 2.83 | 2.70 | 1.80 | 2.40 | 2.67 | 3.00 |
| | 0.16 | **1.63** | 2.77 | 2.63 | 2.53 | 1.80 | 2.40 | 2.53 | 3.00 |
| | 0.10 | **0.93** | 2.83 | 2.93 | 2.83 | 1.37 | 2.63 | 2.80 | 3.00 |
| 0.10 | 0.12 | **1.63** | 2.90 | 2.87 | 2.60 | 1.83 | 2.50 | 2.73 | 3.00 |
| | 0.14 | **1.53** | 2.80 | 2.77 | 2.70 | 1.93 | 2.47 | 2.77 | 3.00 |
| | 0.16 | **1.73** | 2.83 | 2.77 | 2.57 | 1.70 | 2.57 | 2.63 | 3.00 |
| | 0.10 | **0.93** | 2.93 | 2.90 | 2.73 | 1.6 | 2.77 | 2.93 | 3.00 |
| 0.12 | 0.12 | **1.40** | 3.00 | 2.77 | 2.73 | 1.60 | 2.73 | 2.80 | 3.00 |
| | 0.14 | **1.57** | 2.93 | 2.73 | 2.47 | 1.80 | 2.73 | 2.77 | 3.00 |
| | 0.16 | **1.60** | 2.90 | 2.67 | 2.50 | 1.87 | 2.60 | 2.67 | 3.00 |

We use synthetic datasets with hidden community structures to evaluate the performance of our method. We replicate the experiments 30 times under each setting and compare the performance of the error of estimation of the number of communities and the computation time. The average of 30 replications of each setting is recorded in Table A.1 and Table A.2. In these tables, Column **Prop** denotes the subsampling proportion, and column **Pout** denotes the probability of an edge between communities. The rest of the columns are the results of different subsampling methods. The performance of our algorithm is better than that of other methods. As for the results for SBM in Table A.2, the error of estimating the number of communities decreases as the subsampling proportion increases, and the error increases as the observed graphs are noisier (pout increases). As for the computational time recorded in Table A.3 for SBM, we observe that the estimation time of the full sample is two orders of magnitude larger than the time of subsampling and estimation. As the

Table A.2: **SBM**: Error of the estimation of the number of communities for different subsampling methods under different settings.

| Pout | Prop | ORG-sub | MHRW | CSE | FFS | SnowBall | DBN | RW | MDRW |
|------|------|---------|------|-----|-----|----------|-----|-----|------|
|      | 0.10 | **0.47** | 1.70 | 1.93 | 2.00 | 2.33 | 2.00 | 2.00 | 3.00 |
| 0.06 | 0.12 | **0.33** | 1.67 | 1.80 | 1.93 | 2.20 | 2.00 | 2.00 | 3.00 |
|      | 0.14 | **0.10** | 1.77 | 1.77 | 1.93 | 2.13 | 2.00 | 2.00 | 3.00 |
|      | 0.16 | **0.00** | 1.73 | 1.30 | 1.97 | 2.07 | 2.00 | 2.00 | 3.00 |
|      | 0.10 | **0.63** | 1.77 | 1.60 | 1.87 | 2.23 | 2.00 | 2.00 | 3.00 |
| 0.08 | 0.12 | **0.37** | 1.73 | 1.70 | 1.83 | 2.17 | 2.00 | 2.00 | 3.00 |
|      | 0.14 | **0.03** | 1.87 | 1.53 | 1.80 | 2.07 | 2.00 | 2.00 | 3.00 |
|      | 0.16 | **0.07** | 1.83 | 1.67 | 1.83 | 2.03 | 2.00 | 2.00 | 3.00 |
|      | 0.10 | **0.80** | 1.77 | 1.73 | 1.93 | 2.20 | 2.00 | 1.97 | 3.00 |
| 0.10 | 0.12 | **0.33** | 1.73 | 1.67 | 2.00 | 2.17 | 2.00 | 1.93 | 3.00 |
|      | 0.14 | **0.13** | 1.80 | 1.63 | 1.90 | 2.03 | 2.00 | 1.93 | 3.00 |
|      | 0.16 | **0.03** | 1.73 | 1.60 | 1.97 | 2.03 | 2.00 | 1.93 | 3.00 |
|      | 0.10 | **0.73** | 1.67 | 1.83 | 1.87 | 2.07 | 2.00 | 2.00 | 3.00 |
| 0.12 | 0.12 | **0.53** | 1.60 | 1.57 | 1.93 | 2.03 | 2.00 | 2.00 | 3.00 |
|      | 0.14 | **0.33** | 1.63 | 1.53 | 1.93 | 2.00 | 2.00 | 2.00 | 3.00 |
|      | 0.16 | **0.20** | 1.63 | 1.40 | 1.97 | 2.00 | 2.00 | 2.00 | 3.00 |

subsampling proportion increases, the computational time of subsampling methods increases. Though our method's computational time is slightly larger than other subsampling methods, our method has better accuracy.

Table A.3: **SBM**: Comparison of the computation time (s) of the estimation of the number of communities between using the full dataset and the sampled dataset for different subsampling methods under different settings (seconds).

| Pout | Full | Prop | ORG-sub | MHRW | CSE | FFS | SnowBall | DBN | RW | MDRW |
|------|------|------|---------|------|-----|-----|----------|-----|-----|------|
|      |      | 0.10 | 1.11 | 0.42 | 3.27 | 0.43 | 0.44 | 0.29 | 0.37 | 0.28 |
| 0.06 | **22.52** | 0.12 | 1.21 | 0.48 | 3.76 | 0.49 | 0.52 | 0.49 | 0.54 | 0.37 |
|      |      | 0.14 | 1.41 | 0.54 | 4.41 | 0.53 | 0.62 | 0.21 | 0.33 | 0.23 |
|      |      | 0.16 | 1.63 | 0.66 | 5.12 | 0.66 | 0.67 | 0.42 | 0.44 | 0.38 |
|      |      | 0.10 | 1.08 | 0.42 | 3.21 | 0.42 | 0.43 | 0.32 | 0.41 | 0.26 |
| 0.08 | **23.67** | 0.12 | 1.27 | 0.49 | 3.87 | 0.50 | 0.53 | 0.25 | 0.24 | 0.20 |
|      |      | 0.14 | 1.47 | 0.57 | 4.50 | 0.54 | 0.61 | 0.25 | 0.26 | 0.25 |
|      |      | 0.16 | 1.71 | 0.65 | 5.23 | 0.66 | 0.68 | 0.53 | 0.67 | 0.50 |
|      |      | 0.10 | 1.09 | 0.41 | 3.25 | 0.42 | 0.43 | 0.38 | 0.42 | 0.32 |
| 0.10 | **15.67** | 0.12 | 1.30 | 0.49 | 3.91 | 0.50 | 0.53 | 0.27 | 0.28 | 0.17 |
|      |      | 0.14 | 1.52 | 0.56 | 4.57 | 0.54 | 0.61 | 0.26 | 0.34 | 0.26 |
|      |      | 0.16 | 1.78 | 0.65 | 5.28 | 0.66 | 0.68 | 0.57 | 0.68 | 0.52 |
|      |      | 0.10 | 1.13 | 0.42 | 3.28 | 0.42 | 0.45 | 0.18 | 0.24 | 0.22 |
| 0.12 | **19.65** | 0.12 | 1.38 | 0.48 | 4.11 | 0.50 | 0.53 | 0.24 | 0.32 | 0.20 |
|      |      | 0.14 | 1.61 | 0.56 | 4.71 | 0.54 | 0.61 | 0.45 | 0.56 | 0.43 |
|      |      | 0.16 | 1.89 | 0.65 | 5.39 | 0.68 | 0.68 | 0.41 | 0.49 | 0.43 |

## B.2 EXPERIMENTS ON REAL-WORLD DATASETS

### B.2.1 DESCRIPTION OF THE DATASETS

We use five real-world datasets to evaluate the performance of our method as well. These datasets are widely used as benchmark datasets in the study of community detection for graphs. They can be easily found at KONECT, UCI network data repository and Stanford Network Analysis Project Rossi & Ahmed (2015); Rozemberczki et al. (2020a); Sen et al. (2008). All datasets are considered undirected and unweighted graphs, and all self-loop edges and isolated nodes are removed. More details about the datasets are as follow.

*Polblogs*: The political blogs network dataset was collected by Adamic and Glance in 2005. A network is constructed among all the posts published by liberal or conservative bloggers (two communities). Each node represents one post. An edge connects two nodes if one of them is cited by the other.

*Polbooks*: This is a network of books about US politics published around the 2004 presidential election and sold by online bookseller *Amazon.com*. All the books are divided into four communities by NI-LPA. Edges between books represent frequent co-purchasing of books by the same buyers.

*Facebook*: This is an ego-network dataset of the "friend circles" of one anonymous user on Facebook. The network forms friend circles such as family members, high school friends or other friends that are "hand-labeled" by the user.

*Cora*: The Cora dataset describes the citation relationship among scientific publications classified into seven classes. After preprocessing, there remain 2,485 nodes and 5,069 links.

*PubMed*: The PubMed dataset describes the citation relationship among scientific publications classified into seven classes. After preprocessing, there remain 19,717 nodes and 44,338 links.

Table A.4 presents the summary statistics for those real networks we used in this paper. As we can see, these datasets cover different levels of network size, density, number fo communities, clustering coefficient (CC), and imbalance. The degree distributions of all the datasets are presented in Figure A.1.

Table A.4: Key features of the real-world datasets. Node and Edge represent the number of nodes, edges, and communities of the graph, respectively. Density is the edge density, calculated by the ratio of the number of edges in the actual and complete graphs. CC is the clustering coefficient of the whole graph. IM is the imbalance level.

| Dataset | Node | Edge | Density | Community | CC | IM |
|---------|------|------|---------|-----------|------|------|
| Polblogs | 1224 | 16718 | 0.0112 | 2 | 0.2260 | 0.72 |
| Polbooks | 105 | 441 | 0.0400 | 3 | 0.3484 | 0.89 |
| Facebook | 725 | 13030 | 0.0248 | 11 | 0.4576 | 0.71 |
| Cora | 2485 | 5069 | 0.0008 | 7 | 0.0900 | 0.94 |
| PubMed | 19717 | 44338 | 0.0001 | 3 | 0.0537 | 0.96 |

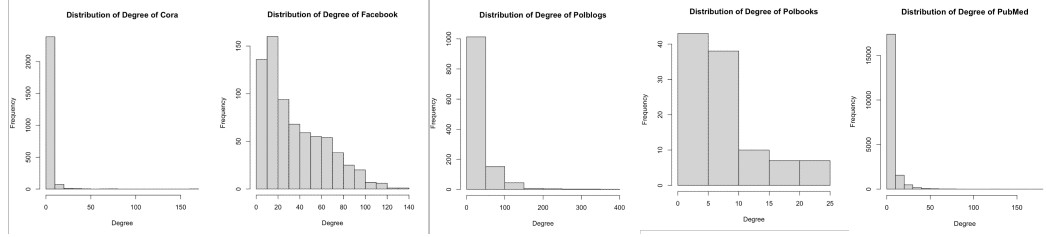

Figure A.1: The degree distribution of the five datasets.

Table A.5: Comparison of the computation time (seconds) of the estimation of the number of communities between using the full dataset and the sampled dataset for each dataset and subsample method with different subsample size.

| Dataset | Full | Ricci | Prop | ORG-sub | MHRW | CSE | FFS | SnowBall | DBN | RW | MDRW |
|---------|------|-------|------|---------|------|-----|-----|----------|-----|-----|------|
| | | | 10% | 0.10 | 0.05 | 0.07 | 0.08 | 0.10 | 0.05 | 0.07 | 0.04 |
| Polbooks | **1.88** | 0.29 | 20% | 0.13 | 0.10 | 0.11 | 0.11 | 0.11 | 0.06 | 0.11 | 0.06 |
| | | | 30% | 0.14 | 0.11 | 0.12 | 0.12 | 0.12 | 0.10 | 0.11 | 0.07 |
| | | | 10% | 0.65 | 0.50 | 0.74 | 0.51 | 0.53 | 0.47 | 0.49 | 0.47 |
| Facebook | **29.14** | 0.71 | 20% | 1.01 | 0.75 | 1.41 | 0.78 | 0.83 | 0.71 | 0.71 | 0.71 |
| | | | 30% | 1.96 | 1.56 | 2.83 | 1.70 | 1.69 | 1.48 | 1.53 | 1.49 |
| | | | 5% | 0.34 | 0.23 | 0.49 | 0.24 | 0.22 | 0.22 | 0.22 | 0.29 |
| Cora | **191.06** | 0.79 | 10% | 0.79 | 0.66 | 1.54 | 0.65 | 0.56 | 0.49 | 0.60 | 0.73 |
| | | | 20% | 2.30 | 1.88 | 4.83 | 1.90 | 1.79 | 1.65 | 1.93 | 2.21 |
| | | | 5% | 0.23 | 0.12 | 0.13 | 0.23 | 0.13 | 0.11 | 0.22 | 0.12 |
| Polblogs | **48.6** | 1.49 | 10% | 0.56 | 0.27 | 0.30 | 0.67 | 0.29 | 0.35 | 0.42 | 0.31 |
| | | | 20% | 1.18 | 0.49 | 0.47 | 1.01 | 0.48 | 0.50 | 0.67 | 0.44 |
| | | | 0.5% | 1.03 | 0.64 | 1.23 | 0.57 | 0.53 | 0.79 | 0.59 | 0.61 |
| PubMed | **NA** | 41.59 | 2% | 4.42 | 3.93 | 11.84 | 4.13 | 3.50 | 2.63 | 3.58 | 3.43 |
| | | | 5% | 8.92 | 8.61 | 43.36 | 8.94 | 8.24 | 6.62 | 8.83 | 7.85 |

### B.2.2 COMPUTATIONAL TIME

The comparisons of errors in estimating the number of communities are presented in the main context. The computation time is also obtained in A.5. Similar to the synthetic datasets, the estimation time of the full sample is two orders of magnitude larger than the time of subsampling (including the time for computing OR curvature for each edge in the graph) and estimation.

The result also shows that the computation time for estimating the number of communities using the subsampling method is much shorter than that using the full dataset. The complexity of all methods is influenced by the size of the graph, especially by the number of nodes. It is no wonder that the computation time of all methods in estimating the number of communities for dataset *Cora* is much longer than other datasets.

### B.3 ADDITIONAL EMPIRICAL RESULTS

In this section, we provide additional empirical results. In real-world networks, the ground truth is not available. We used some data sets that are widely used for community detection tasks, where the communities are labeled with domain knowledge. The manually labeled community structures correspond to different underlying aspects of the nodes. For example, in the Cora citation network (where a node represents a paper), the community label corresponds to the paper's subject (e.g., neural network). We agree that the manually labeled number of communities might not be the ground truth, which is a limitation for real networks.

To overcome this limitation, we also employ another evaluation metric to assess the performance of our methods. In particular, we do not compare our estimation with the manually labeled number of communities. We compare our estimation with the estimation obtained from the full data. These comparison results can be used to evaluate whether the subsampled subgraph can be a good surrogate to carry out computations of interest for the full data. Table A.6 shows the estimation difference between the subsampled and full graphs for four datasets, i.e., Polbooks, Facebook, Cora, and Polblogs. Here we do not show the results for PubMed, since the estimation results of PubMed full graph are not available due to prohibitive computation. As we can see, using this new metric, our method ORG-sub still outperforms other subsampling methods.

Table A.6: The estimation difference between sampling and full.

| Dataset | Prop. | ORG-sub | MHRW | CSE | FFS | SnowBall | DBN | RW | MDRW |
|---|---|---|---|---|---|---|---|---|---|
| | 10% | **0.08** | 0.66 | 0.39 | 2.79 | 1.08 | 0.29 | 0.20 | 1.00 |
| Polbooks | 20% | 0.29 | 0.46 | 0.34 | 0.59 | 0.37 | 0.57 | **0.28** | 0.61 |
| | 30% | **0.13** | 0.75 | 0.34 | 2.86 | 1.11 | 1.80 | 1.70 | 0.99 |
| | 10% | **1.36** | 2.90 | 3.90 | 2.96 | 4.66 | 3.86 | 1.67 | 6.00 |
| Facebook | 20% | **1.31** | 2.84 | 3.73 | 2.24 | 3.09 | 2.87 | 1.44 | 5.94 |
| | 30% | **1.20** | 2.91 | 3.70 | 2.11 | 3.23 | 2.71 | 1.42 | 5.85 |
| | 5% | **0.27** | 0.87 | 0.60 | 2.37 | 1.13 | 0.63 | 2.03 | 0.83 |
| Cora | 10% | 0.80 | 3.90 | 0.47 | 0.80 | **0.60** | 1.60 | 1.13 | 1.33 |
| | 20% | 0.27 | 0.90 | 2.00 | **0.03** | 1.40 | 1.77 | 1.27 | 2.20 |
| | 5% | **0.00** | 1.87 | 0.90 | 2.00 | 0.43 | 1.33 | 1.03 | 0.30 |
| Polblogs | 10% | **0.00** | 0.40 | 0.33 | 0.20 | 0.03 | 0.03 | 0.07 | 0.87 |
| | 20% | **0.00** | 1.87 | 0.90 | 2.00 | 0.43 | 1.33 | 1.03 | 0.30 |

## C DISCUSSION ON EDGE DENSITY

Indeed, the performance of our method depends on the edge density from both theory and empirical results. On one hand, Corollary 4.1.2 basically assumes that $\rho > \frac{log(\tilde{n})}{\tilde{n}}$, where $\tilde{n}$ is the number of sampled nodes, and $\rho$ is the edge density. When $\tilde{n} = 100$, we require $\rho > 0.046$, and when $\tilde{n} = 1000$, we only require $\rho > 0.007$. As the subsample size becomes larger, the constraint imposed on the edge density becomes weaker. On the other hand, we include additional simulation studies to show the empirical performance of our method with different network densities. Particularly, we use the same SBM setting in Section 5.1 to generate synthetic data, except that we let $p_{in} = 0.8\lambda$ and $p_{out} = 0.1\lambda$, where $\lambda \in \{0.2, 0.4, \ldots, 1\}$ controls the edge density level. Here, we fix the sampling proportion as

10%, that is, we sample 90 nodes. Table A.7 shows the edge density and average estimation error under different $\lambda$. As we can see, an increasing edge density comes with a lower error.

Table A.7: **SBM**:Estimation error of the number of communities with different edge density under the SBM model.

| $\lambda$ | 0.2 | 0.4 | 0.6 | 0.8 | 1.0 |
|---|---|---|---|---|---|
| Density | 0.1018 | 0.2034 | 0.3053 | 0.4070 | 0.5085 |
| Error | 2.7 | 2.1 | 1.8 | 1.8 | 1.5 |

## D    RELATED WORK ON GRAPH SUBSAMPLING

Existing graph subsampling techniques can be categorized into three main groups: node-based, edge-based, and exploration-based.

**Node-based sampling.** Random Node (RN) sampling (Leskovec & Faloutsos, 2006) and Degree-based Node sample (DBN) (Leskovec & Faloutsos, 2006) sampling are two most common node-base sampling methods. RN selects a set of nodes uniformly at random from the graph, while DBN selects a node with a probability that is proportional to its degree. DBN has been shown to favor high-degree nodes.

**Edge-based sampling.** Random Edge (RE) sampling (Leskovec & Faloutsos, 2006) generates an induced subgraph by selecting edges uniformly at random. Some variants of RE have been proposed, such as Random Node-Edge (RNE) sampling (Rafiei, 2005) that randomly selects a node and then randomly chooses an adjacent edge. Previous studies (Leskovec & Faloutsos, 2006) have demonstrated that neither RE nor RNE preserves community structures because the resulting sampled graphs are often sparsely connected. Meanwhile, both RE and REN slightly favor high-degree nodes because the probability of selecting a node increases with its degree.

**Exploration-based sampling.** SnowBall sampling (Goodman, 1961) and Forest Fire sampling (FFS) (Leskovec et al., 2005) are two basic exploration-based sampling methods, selecting a fixed fraction of neighbors visited at each iteration. Random walk (RW) (Lovász, 1996) is also a popular exploration-based graph sampling method, which selects the next node at random from the neighbors of the currently selected node. One of the major limitations of RW is that RW is inherently biased towards visiting high-degree nodes. To overcome the drawback of RW, researchers proposed Metropolis Hasting Random Walk Sampler (MHRW) (Hübler et al., 2008) and Multi-dimensional Random Walk Sampler (MDRW) (Ribeiro & Towsley, 2010). Most of the aforementioned literature aims at getting a subgraph that preserves some network summary statistics, such as degree distribution. Only a few works aim at obtaining a subgraph that preserves the community information. (Maiya & Berger-Wolf, 2010) proposed a local greedy search-based community structure expansion sampling (CSE) method to optimize the preservation of community structures.

Despite many successful applications of existing graph sampling methods. They all have various limitations. In particular, node-based and edge-based sampling methods sample nodes or edges independently, ignoring the neighborhoods of seed nodes. They might obtain a disconnected subgraph from a connected graph (Wu et al., 2016). For RW and FF, it has been shown that they could get trapped inside the communities and leave other communities out of the sample entirely (Wu et al., 2016). While the MHRW algorithm ensures that the subgraph preserves degree distribution, the performance of this algorithm is dependent on its sample acceptance ratio. It has been shown that the acceptance ratio of MHRW is typically very low in real-world networks . Therefore, MHRW generally suffers from the sample rejection problem, which clearly degrades the performance of MHRW. For CSE, the selection procedure of the next node is not based on a mathematical framework and one cannot compute the probability of visiting sampled nodes in CSE (Salehi et al., 2012).

## E    EMPIRICAL INVESTIGATION OF FUTURE WORK

Theoretically, the OR curvature has been proven to be related to some network summaries concerning communities, such as the eigenvalues of the graph Laplacian Bauer et al. (2011) and the clustering coefficient (CC) Jost & Liu (2014).

Empirically, we calculated the CC value of the full graph and that of the subsampled graph for two real datasets: Facebook and Cora. We set the subsampling proportion to 10%. We replicated the sampling procedure 100 times and thus obtained 100 subsamples. Using these subsamples, we obtained the 95% confidence interval (CI) by calculating mean $\pm$ 1.96*sd (standard deviation).

For Facebook data, the CI of the CC value is [0.420, 0.537], and the CC value obtained from the full graph is 0.476. For Cora data, the CI is [0.234, 0.290], and the CC value of the full graph is 0.238. As we can see, the 95% CI covers the CC value of the full graph for both real data sets. These observations suggest that our subsampling algorithm could preserve CC to some extent. The consistency theory on CC using our method is under investigation, and the results will be reported in future publications.

