# OpenReview forum: "Subsampling in Large Graphs Using Ricci Curvature"
_ICLR.cc/2023/Conference — ICLR 2023 poster_

### Official Review · Reviewer_yo3s · 2022-10-20

**Confidence:** 3
**Correctness:** 3
**Technical Novelty And Significance:** 3
**Empirical Novelty And Significance:** 2
**Recommendation:** 5

**Clarity, Quality, Novelty And Reproducibility:**


This paper needs to be written more clearly with assumptions stated more prominently and earlier. The idea of basing a subsampling algorithm on theory is good but again what is important is to state what the key assumptions are and then possibly leave the details to an appendix.  The methodology followed is good and there is novelty in using the notions of discrete graph curvature as a guide for what nodes/edges to select for subsampling.  Reproducibility is unclear to this reviewer.

**Strength And Weaknesses:**


Strengths

o The paper addresses subsampling of large graphs to better estimate number of communities.

o The proposed subsampling technique is proved to be asymptotically correct for stochastic block model (SBM) graphs.

Weaknesses

o The number of communities for most graphs, large or small, is not a precise enough number whose accuracy could be measured the way the authors state/measure.

o The writing is not clear/crisp in many important places, some examples are shown below.

o I may have missed this but what is the subsampling ratio used (n~ in the notation of Algorithm 1 at the end of Section 4.1) in Section 5 for synthetic graphs used?  (This is stated to be 10% for real-world graphs which by the way is quite large for large graphs of 50M-1B nodes.)

o No evidence that SBM is representative of real graphs (Twitter, FB, etc)

End of Section 1:
> More importantly, we theoretically show that the estimation of M by the subsampled graph converges to true M.

M, the number of communities in a graph, is not a precise concept in general and there is no ground truth for most graphs for M. So how does one even get a measure of accuracy for the proposed or any other technique in other than synthetic datasets (from the SBM where we start with a set number of communities a priori) and datasets from previous estimates whose accuracy is open to debate?

End of Section 2.1
> The number of communities of the graph G is denoted by M(G)

What is the definition of M(G)?

Beginning of Section 2:
> Without loss of generality, we consider the undirected graph

There is a lot of generality lost if we only allow undirected graphs.  As a matter of fact directed graphs don't even approximate Riemannian manifolds in discrete form since geodesics are always bidirectional. So what generality is *not* lost by considering only undirected graphs?  Why not just say this paper is about undirected graphs only?

End of Section 2:
> In Lemma 3.1, we theoretically prove the lower bound of within-community edges’ OR curvatures are larger than the upper bound of the between-communities edges’ OR curvatures in stochastic block models (SBM).

This, middle of page 4, is the 1st time in the paper that a reference to SBM is given.  So yes, if we define a SBM then we know M exactly but the majority of graphs are not SBMs so the entire theory of this paper is about a special class of graphs. Why not just say so from the beginning?

Middle of Section 5.3
> the community of the graph is more imbalanced.

No clear definition of this concept is provided in any part of this paper other than occasional references to communities with nodes with large degrees as opposed to those without nodes with large degrees.

**Summary Of The Paper:**


The paper is about a novel subsampling technique for large graphs defined in such a way that an important macro feature of large graphs is preserved.  It is stated that one macro feature often not considered is the number of clusters/communities, especially those without 'central' (high-degree) nodes are often left out in prior subsampling techniques.  The main claim here is that the Ricci-Ollivier curvature can help provide a better estimate of the number of communities, especially the marginal communities, ie those without nodes with high enough degrees.  The theoretical basis of the proposed subsampling algorithm (ORG-Sub) is this result: "the ORG-subsampling provides a rigorous theoretical guarantee that the probability of ORG-subsampling taking all communities into the final subgraph converges to one".  This statement is proven and ORG-Sub is applied to some synthetic and real networks to show that it has lower estimation error than prior work.

**Summary Of The Review:**


The paper connects theory to practice via a subsampling method whose asymptotic performance for the SBM is correct as far as number of communities is concerned.  The writing needs to be more crisp and clear.

However, the fundamental notion here is that there is an exact number of communities in any graph -- the ground truth M -- which the proposed algorithm estimates better than other subsampling algorithms. This is debatable in other than limited instances, such as synthetic SBM graphs.  In other words for many real world examples, M, the number of communities is typically very vague so what does it mean for an algorithm to have 1% less error than another?

Lastly, the authors do not provide examples for real 'large graphs' with millions-billions of nodes where subsampling ratio has to be much less than 1%.

---

> ### Author Response · Authors · 2022-11-19
> **Thank you for your comments (1/2)**
>
> We thank the reviewer for the dedicated and insightful review, and please see below for our response to your concerns.
>
> >  ***Comment:** The number of communities for most graphs, large or small, is not a precise enough number whose accuracy could be measured the way the authors state/measure.*
>
> **Response:** Thank you for the comment. Please refer to our general responses **"Clarification on the setting"** to find our responses. In the revised manuscript, we added this discussion in the introduction section.
>
> >  ***Comment:**  I may have missed this but what is the subsampling ratio used (n~ in the notation of Algorithm 1 at the end of Section 4.1) in Section 5 for synthetic graphs used? (This is stated to be 10% for real-world graphs which by the way is quite large for large graphs of 50M-1B nodes.)
>
> **Response:** Thank you for the comment. The subsampling proportion we used for the synthetic dataset is stated in Section 5.1 ({0.1,0.12,0.14,0.16}). For medium-size real-world graphs Polblogs and Cora, we consider 5%, 10%, and 20%. For large-size real-world graph PubMed, we consider 0.5%, 2%, and 5%.
>
> >  ***Comment:** No evidence that SBM is representative of real graphs (Twitter, FB, etc).*
>
> **Response:** Thank you for the comment. Please refer to our general responses **"Universality of SBM"**, where we show evidence that SBM is representative of most real graphs.
>
> >  ***Comment:**  How does one even get a measure of accuracy for the proposed or any other technique in other than synthetic datasets (from the SBM where we start with a set number of communities a priori) and datasets from previous estimates whose accuracy is open to debate?*
>
> **Response:** Thank you for the question. The previous response clarifies the setting in this paper, i.e., there is a ground truth of $M$. In real-world networks,  the ground truth is not available. We used some data sets that are widely used for community detection tasks, where the communities are labeled with domain knowledge. The manually labeled community structures correspond to different underlying aspects of the nodes. For example, in the Cora citation network (where a node represents a paper), the community label corresponds to the paper's subject (e.g., neural network). We agree that the manually labeled number of communities might not be the ground truth, which is a limitation for real networks.
>
>
> To overcome this limitation, we also employ another evaluation metric to assess the performance of our methods. In particular, we do not compare our estimation with the manually labeled number of communities. **We compare our estimation with the estimation obtained from the full data. This comparison result can be used to evaluate whether the sampled subgraph is a good surrogate to carry out computations of interest for the full data.** Table R6 shows the estimation difference between sampling and full for the small-size network (Polbooks and Facebook)  with a sampling proportion of 10\% and the medium-size network (Cora and Polblogs) with a sampling proportion of 5\%. More results under different sampling proportions can be found in Table A.6 in the Appendix. Table R6  does not show the results for PubMed, since the estimation results of PubMed full graph are not available due to prohibitive computation. As we can see from Table R6, using this new metric, our method ORG-sub still outperforms other sampling methods.
>
> Table R6: The estimation difference between sampling and full. Smaller difference indicates better performance.
>
>  | Dataset |  ORG-sub | MHRW | CSE  | FFS  | Snowball | DBN  |  RW  | MDRW |
> |:--------:|:--------:|:----:|:-------:|:----:|:----:|:----:|:--------:|:----:|
> | Polbooks | **0.08** | 0.66 | 0.39 | 2.79 | 1.08 | 0.29 |  0.20 | 1.00 |
> |Facebook | **1.36** | 2.90 | 3.90 | 2.96 | 4.66 | 3.86 | 1.67 | 6.00 |
> |Cora | **0.27** | 0.87 | 0.60 | 2.37 | 1.13 | 0.63 | 2.03 | 0.83 |
> |Polblogs | **0.00** | 1.87 | 0.90 | 2.00 | 0.43 | 1.33 | 1.03 | 0.30 |
>
>
>
> >  ***Comment:** What is the definition of M(G)?*
>
> **Response:** We apologize for the confusion. In Section 1, we first present  $M$ as a general notation for the true number of communities. At the end of Section 2.1, we define  $M(G)$ as the true number of communities for specific graph $G$. To avoid confusion, we remove the redundant notation $M(G)$ and keep using $M$ as the true number of communities.
>
> >  ***Comment:** As a matter of fact directed graphs don't even approximate Riemannian manifolds in discrete form since geodesics are always bidirectional. So what generality is not lost by considering only undirected graphs? Why not just say this paper is about undirected graphs only?*
>
> **Response:** Thank you for the comment and suggestion. We revised the statement at the beginning of Section 2 as follows.
>
> *"Considering that directed graphs don't approximate Riemannian manifolds in discrete form since geodesics are always bidirectional, we only focus on undirected graphs in this paper."*

---

> > ### Comment · Reviewer_yo3s · 2022-11-21
> > **Acknowledgment**
> >
> > I acknowledge and thank the authors for their clarifying comments.

---

> > > ### Author Response · Authors · 2022-11-22
> > > **Kind request for re-evaluation or further discussion**
> > >
> > > Dear reviewer,
> > >
> > > We highly appreciate you going through our responses to your comments.
> > >
> > > We are wondering whether our response addressed all your concerns. If not, we are happy to have further discussions. If yes, could your please reconsider your evaluation or provide additional feedback for the paper?
> > >
> > > Many thanks in advance for your time and efforts!

---

> > > > ### Comment · Reviewer_yo3s · 2022-11-22
> > > > **Re: Kind request for re-evaluation or further discussion**
> > > >
> > > > As I stated in my review, and despite the authors' response which consisted of just citing earlier claims about SBMs, I don't see strong enough evidence (in this work or in prior work) that demonstrates SBMs are universal models for real-life networks and more pertinently that there can be an exact enough count of clusters which may be measured with the accuracy the authors claim in this paper and in their various benchmarking tables.  Yes, this can be done with synesthetic data where the number of clusters is set a priori but this is not how real data works.

---

> > > > > ### Author Response · Authors · 2022-11-30
> > > > > **Thank you for your new comments (1/2)**
> > > > >
> > > > > Thank you for the comment. We agree with the reviewer that SBMs are not universal for all real-life networks. However, as  George E.P. Box pointed out, all models are wrong, but some are useful. SBMs are very useful in many applications from both theoretical and practical views, as we show below.  Note that SBMs refer to SBM and its variants such as degree-correlated SBM (DCSBM).
> > > > >
> > > > > First, from a theoretical perspective, SBMs provide a precise way for researchers to model the randomness in the network data (Rohe et al., 2011). SBM-based community detection methods (Young et al., 2018; Zhang et al., 2018; Abbe, 2017; Zhao et al., 2012; Cohen-Addad et al., 2020; Wang et al., 2021b) usually fit SBMs to data and find the parameters that maximize the likelihood, thus can be viewed as a parametric community detection way.  Actually, another popular line of community detection research is in a nonparametric way, i.e., finding a partition maximizing some objective function that is defined by heuristics or insights, such as Girvan–Newman algorithm (Girvan & Newman, 2006), spectral clustering (Newman, 2013), modularity maximization (Newman & Girvan, 2004; Blondel et al., 2008) and nonnegative matrix factorization (Lu et al., 2020).
> > > > >
> > > > > Recent studies (Young et al., 2018) show that the likelihood functions of SBMs are equivalent to the objective functions in most of the aforementioned nonparametric algorithms. For example, maximizing the likelihood of SBMs has been proven to be equivalent to maximizing (1)  modularity (Zhang & Moore, 2014; Newman, 2016; Pamfil et al., 2019; Veldt et al., 2018), (2) the objective function in nonnegative matrix factorization algorithm (Zhang et al., 2018), (3) the objective function in spectral clustering (Young et al., 2018; Newman, 2013). **In summary, many community detection methods can be reformulated under the framework of SBMs with sound statistical foundations.**
> > > > >
> > > > > Second, from a practical perspective, SBMs have been successfully applied to model many real networks. For example, SBMs have shown superior performance in detecting community structure in social networks, biological networks, and brain networks (Abbe, 2017; Girvan & Newman, 2022; Newman, 2013; Newman & Girvan, 2004; Blondel et al., 2008; Lu et al., 2020). In addition to community detection, SBMs have also been successfully applied to predict novel links in protein-protein networks (Valles-Catala et al., 2016; Wang et al., 2022), detect change points in political social networks (Bao & Michailidis, 2018; Karrer & Newman, 2011), brain networks (Peixoto, 2018; Faskowitz et al., 2018). **This evidence validates the practical usefulness of SBMs.**
> > > > >
> > > > > Indeed, we agree with the reviewer that it is hard to know the true number of communities or community structures for real-life networks, e.g., we can get a partition based on gender or another partition based on age.  This is the limitation of all community detection literature when evaluating performance in real networks. To address this limitation, we made efforts in two aspects. (1) We used well-studied example networks, e.g., karate clubs in Zachary (1977), political blogs (Adamic & Glance, 2005), college football (Girvan & Newman, 2002) with widely agreed community divisions based on the domain knowledge and on consensus derived from repeated analysis with many different community detection algorithms. (2) We added another metric for real networks (results can be found in Table R6) in our previous response. We hope that the above response can adequately address the reviewer's concerns.

---

> > > > > > ### Author Response · Authors · 2022-11-30
> > > > > > **Thank you for your new comments (2/2)**
> > > > > >
> > > > > > References:
> > > > > >
> > > > > > [1] Karl Rohe, Sourav Chatterjee, and Bin Yu. Spectral clustering and the high-dimensional stochastic blockmodel. *The Annals of Statistics*, 39(4):1878–1915, 2011.
> > > > > >
> > > > > > [2] Jean-Gabriel Young, Guillaume St-Onge, Patrick Desrosiers, and Louis J Dubé. Universality of the stochastic block model. *Physical Review E*, 98(3):032309, 2018.
> > > > > >
> > > > > > [3] Zhong-Yuan Zhang, Yujie Gai, Yu-Fei Wang, Hui-Min Cheng, and Xin Liu. On equivalence of likelihood maximization of stochastic block model and constrained nonnegative matrix factorization. *Physica A: Statistical Mechanics and its Applications*, 503:687–697, 2018.
> > > > > >
> > > > > > [4] Emmanuel Abbe. Community detection and stochastic block models: recent developments. *The Journal of Machine Learning Research*, 18(1):6446–6531, 2017.
> > > > > >
> > > > > > [5] Yunpeng Zhao, Elizaveta Levina, and Ji Zhu. Consistency of community detection in networks under degree-corrected stochastic block models. *The Annals of Statistics*, 40(4):2266–2292, 2012.
> > > > > >
> > > > > > [6] Michelle Girvan and Mark EJ Newman. Community structure in social and biological networks. *Proceedings of the national academy of sciences*, 99(12):7821–7826, 2002.
> > > > > >
> > > > > > [7] Mark EJ Newman. Spectral methods for community detection and graph partitioning. *Physical Review E*, 88(4):042822, 2013.
> > > > > >
> > > > > > [8] Mark EJ Newman and Michelle Girvan. Finding and evaluating community structure in networks. *Physical review E*, 69(2):026113, 2004.
> > > > > >
> > > > > > [9] Vincent D Blondel, Jean-Loup Guillaume, Renaud Lambiotte, and Etienne Lefebvre. Fast unfolding of communities in large networks. *Journal of statistical mechanics: theory and experiment*, 2008 (10):P10008, 2008.
> > > > > >
> > > > > > [10] Hong Lu, Xiaoshuang Sang, Qinghua Zhao, and Jianfeng Lu. Community detection algorithm based on nonnegative matrix factorization and pairwise constraints. *Physica A: Statistical Mechanics and its Applications*, 545:123491, 2020.
> > > > > >
> > > > > > [11] Pan Zhang and Cristopher Moore. Scalable detection of statistically significant communities and hierarchies, using message passing for modularity. *Proceedings of the National Academy of Sciences*, 111(51):18144–18149, 2014.
> > > > > >
> > > > > > [12] Mark EJ Newman. Equivalence between modularity optimization and maximum likelihood methods for community detection. *Physical Review E*, 94(5):052315, 2016.
> > > > > >
> > > > > > [13] A Roxana Pamfil, Sam D Howison, Renaud Lambiotte, and Mason A Porter. Relating modularity maximization and stochastic block models in multilayer networks. *SIAM Journal on Mathematics of Data Science*, 1(4):667–698, 2019.
> > > > > >
> > > > > > [14] Nate Veldt, David F Gleich, and Anthony Wirth. A correlation clustering framework for community detection. *In Proceedings of the 2018 World Wide Web Conference*, pp. 439–448, 2018.
> > > > > >
> > > > > > [15]Wei Bao and George Michailidis. Core community structure recovery and phase transition detection in temporally evolving networks. *Scientific reports*, 8(1):1–16, 2018.
> > > > > >
> > > > > > [16] Brian Karrer and Mark EJ Newman. Stochastic blockmodels and community structure in networks. *Physical review E*, 83(1):016107, 2011.
> > > > > >
> > > > > > [17] Tiago P Peixoto. Nonparametric weighted stochastic block models. *Physical Review E*, 97(1):012306, 2018.
> > > > > >
> > > > > > [18] Joshua Faskowitz, Xiaoran Yan, Xi-Nian Zuo, and Olaf Sporns. Weighted stochastic block models of the human connectome across the life span. *Scientific reports*, 8(1):1–16, 2018.
> > > > > >
> > > > > > [19] Vincent Cohen-Addad, Adrian Kosowski, Frederik Mallmann-Trenn, and David Saulpic. On the power of louvain in the stochastic block model. *Advances in Neural Information Processing Systems*, 33:4055–4066, 2020.
> > > > > >
> > > > > > [20] Peng Wang, Huikang Liu, Zirui Zhou, and Anthony Man-Cho So. Optimal non-convex exact recovery in stochastic block model via projected power method. In International Conference on Machine Learning, pp. 10828–10838. *PMLR*, 2021b.
> > > > > >
> > > > > > [21] Toni Valles-Catala, Francesco A Massucci, Roger Guimera, and Marta Sales-Pardo. Multilayer stochastic block models reveal the multilayer structure of complex networks. *Physical Review X*, 6 (1):011036, 2016.
> > > > > >
> > > > > > [22] Xiaojuan Wang, Wen Yang, Yue Yang, Yizhou He, Jun Zhang, Lusheng Wang, and Lun Hu. Ppisb: a novel network-based algorithm of predicting protein-protein interactions with mixed membership stochastic blockmodel. *IEEE/ACM Transactions on Computational Biology and Bioinformatics*, 2022.
> > > > > >
> > > > > > [23] Wayne W Zachary. An information flow model for conflict and fission in small groups. *Journal of anthropological research*, 33(4):452–473, 1977.
> > > > > >
> > > > > > [24] Lada A Adamic and Natalie Glance. The political blogosphere and the 2004 us election: divided they blog. *In Proceedings of the 3rd international workshop on Link discovery*, pp. 36–43, 2005.
> > > > > >
> > > > > > [25] Michelle Girvan and Mark EJ Newman. Community structure in social and biological networks. *Proceedings of the national academy of sciences*, 99(12):7821–7826, 2002.

---

> ### Author Response · Authors · 2022-11-19
> **Thank you for your comments (2/2)**
>
> >  ***Comment:**  Middle of page 4, is the 1st time in the paper that a reference to SBM is given. So yes, if we define a SBM then we know M exactly but the majority of graphs are not SBMs so the entire theory of this paper is about a special class of graphs. Why not just say so from the beginning?*
>
> **Response:**  Thank you for the comment. Please refer to our general responses **"Clarification on the setting"** and **"Universality of SBM"** to find our responses. In the revised manuscript, we followed your suggestion to make the above clarification in the introduction section.
>
> >  ***Comment:** Middle of Section 5.3: No clear definition of this concept is provided in any part of this paper other than occasional references to communities with nodes with large degrees as opposed to those without nodes with large degrees.*
>
> **Response:** Thank you for the comment. In the Middle of Section 5.3, we revised the statement to present the definition of imbalance.
>
> *"Let $n_1, \ldots, n_M$ denote the number of nodes in $M$ communities. Here, we calculate the normalized Shannon Entropy of $n_1, \ldots, n_M$ as a metric to measure the imbalance level, i.e., we calculate  $\mbox{IM} = -\frac{1}{\log M}\sum_{i=1}^{M} \frac{n_{i}}{n} \log \frac{n_{i}}{n}$  and  a higher $\mbox{IM}$ value means that the communities are more imbalanced.  Table A.4 in the appendix summarizes the $\mbox{IM}$"*
>
> >  ***Comment:** The authors do not provide examples for real 'large graphs' with millions-billions of nodes where subsampling ratio has to be much less than 1%.*
>
> **Response:** Following your suggestion,  we added one more real-world example, i.e., the PubMed citation network with 19,717  nodes and  44,338 edges. The network density is only 0.0001; thus, PubMed is a very sparse network. Please refer to Table R1 in our general responses **"Additional large real network"** to find the results of PubMed. As we can see, our method ORG-sub still outperforms other methods.
>
>
> >  ***Comment:** Reproducibility is unclear.*
>
> **Response:** Thank you for the comment. Our code and data are provided in the updated supplementary material.

---

### Official Review · Reviewer_5vqt · 2022-10-24

**Confidence:** 3
**Clarity, Quality, Novelty And Reproducibility:** 1. It would be clearer if you could m…
**Correctness:** 3
**Technical Novelty And Significance:** 2
**Empirical Novelty And Significance:** 2
**Recommendation:** 5

**Strength And Weaknesses:**

Strengths:
1. This proposed method is based on a theoretically sound analysis of the gap between OR curvatures for within-community and cross-community edges.
2. Empirical results demonstrate a superior performance of OR curvature gradient-based subsampling methods.
3. Like random walk methods, we can pre-compute OR curvature statistics for fast sampling.

Weaknesses:
1. I'm not fully convinced that OR curvature-based method is suitable for massive graphs like Twitter since those graphs are highly sparse and exhibit significantly different properties than SBM.
2. There is no theoretical evidence to show that the proposed OR curvature-based graph subsampling is better than previous methods.
3. Empirical results are weak from my point of view (see Clarity part for details.)

**Summary Of The Paper:**

This paper introduces Ollivier Ricci (OR) curvature to distinguish within-community edges and cross-community edges. OR curvature uses Wasserstein distance and geodesic distance, where the Wasserstein distance can be computed by Sinkhorn algorithms. Given the nice properties of OR curvature on stochastic block models (SBM), the paper proposes a sampling algorithm by incrementally selecting neiborhood edges with the highest OR curvature gradients. The sampling algorithm is guaranteed to preserve minor communities better, thus better estimating the total number of communities compared to existing methods, including random walk methods.

**Summary Of The Review:**

This paper proposes a theoretically sound subsampling method for graphs using OR curvature gradient. From my point of view, the paper still has room to improve its theory and experiments.

---

> ### Author Response · Authors · 2022-11-19
> **Thank you for your comments (1/2)**
>
> Thank you for the dedicated review and helpful feedback. We tried to address all of your comments in our response and the updated paper.
>
> >  ***Comment:** I'm not fully convinced that OR curvature-based method is suitable for massive graphs like Twitter since those graphs are highly sparse and exhibit significantly different properties than SBM.*
>
> **Response:** Thank you for the comment. First, please refer to our general responses **"Universality of SBM"**, where we show evidence that SBM is representative of most real graphs.
>
> Second, we empirically show that our method is suitable for the large sparse network. We added a large sparse network, i.e., PubMed citation network. PubMed has 19,717  nodes and  44,338 edges. The network density is only 0.0001, which is highly sparse. Please refer to Table R1 in our general responses **"Additional large real network"** to find the results of PubMed. As we can see, our method ORG-sub still outperforms other methods.
>
>
> >  ***Comment:** There is no theoretical evidence to show that the proposed OR curvature-based graph subsampling is better than previous methods.*
>
> **Response:**  We highly appreciate the comment. To the best of our knowledge,  theorems regarding the number of communities are still lacking in existing graph sampling literature. To bridge the long-lasting gap, we establish the theorems in this paper.
>
> To better answer your question, we provided the theoretical property of an existing popular graph sampling method, i.e., random walk (Lovász, 1996). In Corollary 4.1.4 in the updated manuscript, we theoretically present the advantage of ORG-sub over the random walk method. **In particular, under the assumptions in Theorem 4.1, we prove that $P(v_{RW}\in B_i)< P(v\in B_i)$**, where $P(v_{RW}\in B_i)$ represents the probability of sampling a node from community $B_i$, $i=1,\ldots, M$ using random walk, and $P(v\in B_i)$  represents the probability using our method ORG-sub. The proofs can be found in the Appendix (on pages 20 and 21). This means the ORG-sub is more likely to obtain a subsample containing nodes in different communities.
>
>
> In addition to the theoretical advantage over the random walk, we added a section, "Related Work on graph sampling", in the Appendix to discuss the limitations of existing graph sampling methods. The limitations can be summarized as follows.
>
> *"Despite many successful applications of existing graph sampling methods, they have some limitations. In particular, node-based and edge-based sampling methods sample nodes or edges independently, ignoring the neighborhoods of seed nodes. They might obtain a disconnected subgraph from a connected graph (Wu et al., 2016). For random walk and Forest Fire sampling, it has been shown that they could get trapped inside the communities and leave other communities entirely out of the sample  (Wu et al., 2016). While the Metropolis Hasting Random Walk Sampler (MHRW) algorithm ensures that the subgraph preserves degree distribution,  the performance of this algorithm is dependent on its sample acceptance ratio. It has been shown that the acceptance ratio of MHRW is typically very low in real-world networks (Bar-Yossef et al., 2000). Therefore, MHRW generally suffers from the low-efficiency sampling problem, which degrades the performance of MHRW. For community structure expansion sampling (CSE),  the selection procedure of the next node is not based on a mathematical framework, and one cannot compute the probability of visiting sampled nodes in CSE (Salehi et al., 2012)."*

---

> ### Author Response · Authors · 2022-11-19
> **Thank you for your comments (2/2)**
>
> >  ***Comment:** Empirical results could be more persuasive if you include the following: (i) Dataset statistics (# nodes, # edges, degree distribution, etc.). (ii) How you select hyper-parameters, e.g., alpha. (iii) A broader range in the proportion of data you use and the scaling of computation cost.*
>
>
> **Response:** Thank you for the comments. We reply to your comments one by one.
>
> (i) Table A.4 in the appendix presents the summary statistics for real networks, including the number of nodes, number of edges, and edge density. We further present the degree distribution of these networks in Figure 6.  As we can see, the networks we considered span a wide range of network sizes (#nodes ranging from 102 to 19,717) and network densities (ranging from 0.0001 to 0.0416).
>
> (ii) We set $\alpha=0.5$ since it is a common setup suggested by  Ni et al. (2018); Ye et al. (2019). Here, $\alpha$  is used to keep the probability mass of $\alpha$ at node $u$ itself and distribute the rest uniformly over the neighborhood, and $\alpha=0.5$  means that each node keeps 50\% of the probability mass to itself.
>
> (iii) We followed your suggestion and considered a broader range of the proportion. In particular, for small-size networks Polbooks and Facebook, we consider sampling proportions 10\%, 20\% and  30\%. For medium-size networks Polblogs and Cora, we consider 5\%, 10\%, and 20\%. For large-size network PubMed, we consider 0.5\%, 2\%, and 5\%.
>
> >  ***Comment:** It would be clearer if you could make a table to compare the time complexity between ORG-sub and other methods.*
>
> **Response:** Following your suggestion, we compared the time complexity between ORG-sub and other methods. Please refer to our general responses **"Computational Time"** to find the results.
>
>
>
> References:
> [1] Jure Leskovec and Christos Faloutsos. Sampling from large graphs. In *Proceedings of the 12th ACM SIGKDD international conference on Knowledge discovery and data mining*, pp. 631–636, 2006.
>
> [2] Davood Rafiei. Effectively visualizing large networks through sampling. In *VIS 05. IEEE Visualization,* 2005., pp. 375–382. IEEE, 2005.
>
> [3] Sang Hoon Lee, Pan-Jun Kim, and Hawoong Jeong. Statistical properties of sampled networks. *Physical review E*, 73(1):016102, 2006.
>
> [4] Jure Leskovec, Jon Kleinberg, and Christos Faloutsos. Graphs over time: densification laws, shrinking diameters and possible explanations. In *Proceedings of the eleventh ACM SIGKDD international conference on Knowledge discovery in data mining*, pp. 177–187, 2005.
>
> [5] László Lovász. Random walks on graphs: A survey in combinatorics. *Bolyai Society Mathematical Studies*, pp. 353–397, 1996.
>
> [6] Yanhong Wu, Nan Cao, Daniel Archambault, Qiaomu Shen, Huamin Qu, and Weiwei Cui. Evaluation of graph sampling: A visualization perspective. *IEEE transactions on visualization and computer graphics*, 23(1):401–410, 2016.
>
> [7] Chien-Chun Ni, Yu-Yao Lin, Jie Gao, and Xianfeng Gu. Network alignment by discrete ollivier-ricci flow. In *International Symposium on Graph Drawing and Network Visualization*, pp. 447–462. Springer, 2018.
>
> [8] Ze Ye, Kin Sum Liu, Tengfei Ma, Jie Gao, and Chao Chen. Curvature graph network. In *International Conference on Learning Representations*, 2019.

---

> ### Author Response · Authors · 2022-11-28
> **Kind request for re-evaluation or further discussion**
>
> Dear reviewer
>
> Thank you again for having taken the time to evaluate our submitted manuscript thoroughly.
>
> We would very much appreciate you going through our responses to your comments, as well as the significantly updated manuscript in response to them.
>
> We are wondering whether our response addressed all your concerns. If not, we are happy to have further discussions. If yes, could your please reconsider your evaluation or provide additional feedback for the paper?
>
> Many thanks in advance for your time and efforts.

---

> ### Author Response · Authors · 2022-12-11
> **Kind request for feedback**
>
> Dear Reviewer,
>
> Thank you again for your review. We have followed your suggestions by providing a theoretical guarantee to show that the proposed method is better than the baseline method and adding more experiment results.
>
> As the discussion period is ending soon, we would greatly appreciate it if you could provide feedback on our response. We will be happy to address any remaining concerns.
>
> Best,
> Authors

---

### Official Review · Reviewer_FvBB · 2022-10-24

**Confidence:** 5
**Correctness:** 4
**Technical Novelty And Significance:** 4
**Empirical Novelty And Significance:** 4
**Recommendation:** 8

**Clarity, Quality, Novelty And Reproducibility:**

Clarity. The paper is clear (of course the proofs are less clear due to the algebraic complexity even for a simple SBM model".
Quality. High quality provided that convincing arguments about efficiency are clearly exposed in the discussion.
Novelty. Same as quality. In addition, this is comparable to greedy community detectors but more principled.
Reproducibility. Code not yet been released. Please do.

**Details Of Ethics Concerns:**

Ok

**Strength And Weaknesses:**

* Strengths. The curvature idea is very nice and inspiring. It has been also used for quantifying bottlenecks in graphs and their impact in the performance of GNNs. The theorems are very nice given the relative simplicity of the SBM model.

* Weaknesses. In the negative part, the claim that the complexity of the algorithm is close to O($|E|$) is not well documented. I followed the literature and I cannot find any statement about the linearity of the Sinkhorn algorithm unless a significant distortion of the embedding. Please give convincing arguments that computational time is not an issue herein.

**Summary Of The Paper:**

Detailed considerations:

Main concern. How to access curvature without knowing the full graph? It is not clear how to compute the LINEAR SINKHORN ALGORITHM. Should we compute the Wasserstein distance between any node in the graphs as stated in Algorithm-1? Please clarify this:

Please see "Sinkhorn Distances: Lightspeed Computation of Optimal Transport" where they state:

"For a general matrix M, the worst case complexity of computing that optimum scales in $O(d^3 \log d)$” where d is the size of the matrix (graph).

“In the particular case that the metric probability space of interest can be embedded in R n and n is small, computing or approximating optimal transport distances can become reasonably cheap. Indeed, when n = 1, their computation only requires O(d log d) operations. When n ≥ 2, embeddings of measures can be used to approximate them in linear time (Indyk and Thaper, 2003; Grauman and Darrell, 2004; Shirdhonkar and Jacobs, 2008) and network simplex solvers can be modified to run in quadratic time (Gudmundsson et al., 2007; Ling and Okada, 2007). However, the distortions of such embeddings (Naor and Schechtman, 2007) as well as the exponential increase of costs incurred by such modifications as n grows make these approaches inapplicable when n exceeds 4” where n is the embedding dimension

And later:
Such algorithms include Sinkhorn’s celebrated fixed point iteration (1967), which is known to have a linear convergence (Franklin and Lorenz, 1989; Knight, 2008). Unlike other iterative simplex-like methods that need to cycle through complex conditional statements, the execution of Sinkhorn’s algorithm only relies on matrix-vector products

In section 4.3, it is argued that “Aside from the computation complexity of the OR curvature of the graph (which is near-linear by Sinkhorn’s algorithm) “. Even if this is true (I cannot infer that from the “Sinkhorn Distances paper”, the complexity is n|E| which means O($n^2$)  or O($n^3$) for a dense graph.

If this is correct,  cubic complexity is equivalent to eigenvalues/vector computation. An interesting conclusion is that by computing the Laplacian pseudo inverse we can have a similar algorithm based on the commute times or effective resistances (which also satisfy that they are smaller inside the community than between communities). In any case, for a large graph, an O($n^3$) complexity seems unavoidable.

Although more details on “computational time” are placed in the Appendix, we can only find these statements “As for the computational time, we observe that the estimation time of the full sample is two orders of magnitude larger than the time of subsampling and estimation. As the subsampling proportion increases, the computational time of subsampling methods increases. Though our method’s computational time is slightly larger than our subsampling methods, our method has better accuracy.“ Please clarify, since knowing the averaged computational time is critical when dealing with large graphs as claimed when motivating the approach.


Second: The theory of Ricci Curvature is not novel, only the concept of gradient bounds for inter and intra-community edges. The convergence theorems are nice (use of Matrix Chernoff bounds for instance) and the simplicity of the SBM model contributes to finding interpretable bounds in terms of probabilities p_{in} and p_{out}

Third: Results are ok for a limited set of datasets.

**Summary Of The Review:**

The paper is well-motivated and the theoretical background is sound. However, there is a weak argumentation regarding the computational complexity of the algorithm. I consider that this is a fundamental issue when dealing with very large graphs as suggested in the motivation.

---

> ### Author Response · Authors · 2022-11-19
> **Thank you for your comments (1/2)**
>
> We thank the reviewer for the positive and insightful comments. Please see below for our response to your concerns.
>
>
> >  ***Comment:** How to access curvature without knowing the full graph?*
>
> **Response:** Thank you for the comment. The problem setup in this paper is that we have the full large graph, but it is computationally prohibitive to estimate the number of communities using this full graph. For example,  estimating the number of communities requires $O(hn^3)$ computational cost (Li et al., 2020), where $n$ is the number of nodes, and $h$ is the number of choices of the number of communities. The computational cost is unaffordable when there are thousands or millions of nodes. Thus, we aim to get a  subgraph with $\tilde{n}<<n$ nodes,  such that we can use it to get an accurate estimation while reducing the computational cost to  $O(h\tilde{n}^3)$.
>
> Indeed, there is another scenario where the full graph is unknown, and the target population is hidden, e.g., drug abusers in urban areas. In this case, researchers may employ certain sampling algorithms on the graph to explore the hidden population (Hu & Lau, 2013). This unknown full-graph scenario is beyond the scope of this paper. In the future, it will be of our interest to develop a novel sampling strategy for this scenario. We will add the above discussion in  Section 6.
>
> Note that even though we focus on the setting where the full graph is known, we do not necessarily need to calculate the curvature for all edges in the whole graph, as shown in the next response, where we present a fast version of our algorithm.
>
>
> >  ***Comment:** Please give convincing arguments that computational time is not an issue herein.*
>
> **Response:**
> Thank you for your constructive comments.
>
> First, we would like to clarify the computation of the Sinkhorn algorithm. We agree with the reviewer that the worst-case complexity of computing that optimum scales in $O(n^3\log n)$ where $n$ is the size of the adjacency matrix (graph). However, the computational complexity of Sinkhorn is not $O(n^3\log n)$. As a matter of fact, at the end of Section 5 in Cuturi (2013), the authors show that the empirical complexity of the Sinkhorn algorithm is $O(n^2)$. **In this paper, we employed a recent popular approximation method (Altschuler et al., 2019)  to further accelerate the computation of Sinkhorn.** In Altschuler et al. (2019), the authors combine the Nystrom method and Sinkhorn scaling to reduce the computational cost to linear order **$O(n)$**. The evidence can be found in the middle of Section 5 in Altschuler et al. (2019), where they stated, "the running time of NYS-SINK is empirically well-approximated by a line with slope 1 in the log-log plane – representing a complexity of $O(n)$ – whereas the running time of SINKHORN scales as $O(n^2)$". In summary, the algorithm we used to calculate the transportation distance is linear in $n$.
>
>
> Second, we would like to clarify the complexity of computing OR curvature. The computational cost for calculating the OR curvature for all edges is $O(n|E|)$. We also agree with the reviewer that the complexity means $O(n^2)$ for a sparse graph and  $O(n^3)$ for a dense graph. However, as reviewer Rvtn and reviewer 5vqt suggested, **most large graphs are sparse**. This conclusion is also supported by (Kunszenti-Kovács et al., 2019; Goswami et al., 2018) and by the numerical density of the real data in this paper, as shown in Table A.4. Particularly, the numerical density ranges from 0.0001 to 0.0400. In addition, we can further accelerate the computation by implementing parallel computation. Suppose we use $R$  cores; the complexity for OR curvature of all edges becomes $O(\frac{n|E|}{R})$. Table R4 shows the computation time for the curvatures with a 40-core processor. Here, we considered an additional large network PubMed with  19,717 nodes (details of these networks can be found in our general responses "Additional large real networks"). **As we can see, the calculation of   OR curvature  is  feasible.**
>
> Table R4:   Curvature computation time for different datasets.
>
> |  Dataset  |  Polblogs | Polbooks | Facebook | Cora | PubMed |
> | :---   |    :----:   |    :----: | :----:   |    :----:  | :----:   |
> | Time (s) |  1.49 |0.29  |  0.71 |  0.79 | 41.59 |

---

> ### Author Response · Authors · 2022-11-19
> **Thank you for your comments (2/2)**
>
> -----Continuing part 1-----
>
> Third, we propose a **fast version** of our algorithm for a very large or dense graph called **ORG-sub1**. The basic idea of ORG-sub1 is that we do not need to calculate the OR curvature of all edges, and **we only need to calculate the OR curvature of neighboring edges of the selected nodes**. In particular, ORG-sub1 works analogously to  Algorithm 1 (as shown in the paper). The only modification is that   ORG-sub1 does not require the input of OR curvature of all edges,  and ORG-sub1 calculates the curvature of neighboring edges in "Step 1". The expected number of edges that need the calculation of curvature is $\tilde{n}\bar{d}$, where $\bar{d}$ is the average degree. In this case,  the curvature calculation complexity is $O(n\tilde{n}\bar{d})$. Since $\bar{d}=\frac{2|E|}{n}$, we have  $n\tilde{n}\bar{d}=2\tilde{n}|E|$. Thus the complexity is rewritten as $O(\tilde{n}|E|)$. By using parallel computation, the complexity can be further reduced to $O( \frac{\tilde{n}|E|}{R})$. When fixing $\tilde{n}$ to a small number, the  complexity is $O( \frac{|E|}{R})$. **For sparse networks, the complexity is only $O(\frac{n}{R})$**, and even for the dense network, which is rare,  the complexity is only  $O(\frac{n^2}{R})$.
>
> Empirically, we applied ORG-sub1 to the real networks using a  40-core processor. The computational time for a subgraph with  100 nodes is reported in Table R5, from which one can conclude that even for large networks such as PubMed,  the calculation of curvature is not an issu
>
> Table R5: Averaged curvature computational  time for different datasets using ORG-sub1.
>
> |  Dataset  |  Polblogs | Polbooks | Facebook | Cora | PubMed |
> | :---   |    :----:   |    :----: | :----:   |    :----: | :----:   |
> | Time (s) |  0.98 |0.25  |  0.69 |  0.75 | 1.46 |
>
>
> In practice, when the network size is very large, such as millions of nodes, or very dense, we suggest using ORG-sub1; otherwise, we suggest using  ORG-sub. The advantage of ORG-sub is that it only needs to compute all edges' curvature once, and the values can be reused to obtain different subsamples with different sampling proportions. We will add the above discussion in the appendix.
>
>
> >  ***Comment:** Please clarify, since knowing the averaged computational time is critical when dealing with large graphs as claimed when motivating the approach.*
>
> **Response:** Thank you for the comment. Please refer to our general responses **"Computational Time"**, where we compare the computational time of different methods.
>
> >  ***Comment:** Reproducibility. Code not yet been released. Please do.*
>
> **Response:** Thank you for the suggestion. Our code and data are provided in the updated supplementary material.
>
> References:
> [1] Marco Cuturi. Sinkhorn distances: Lightspeed computation of optimal transport. *Advances in neural information processing systems*, 26, 2013.
>
> [2] Jason Altschuler, Francis Bach, Alessandro Rudi, and Jonathan Niles-Weed. Massively scalable sinkhorn distances via the nyström method. *Advances in neural information processing systems*, 32, 2019.
>
> [3] Dávid Kunszenti-Kovács, László Lovász, and Balázs Szegedy. Measures on the square as sparse graph limits. *Journal of Combinatorial Theory*, Series B, 138:1–40, 2019.
>
> [4] Swati Goswami, CA Murthy, and Asit K Das. Sparsity measure of a network graph: Gini index. *Information Sciences*, 462:16–39, 2018.
>
> [5] Tianxi Li, Elizaveta Levina, and Ji Zhu. Network cross-validation by edge sampling. *Biometrika*, 107(2):257–276, 2020

---

> > ### Comment · Reviewer_FvBB · 2022-11-21
> > **Acceptance**
> >
> > Ok, I lean towards acceptance. Thank you for your clarifications.

---

> > > ### Author Response · Authors · 2022-11-22
> > > **Thanks you for your reply**
> > >
> > > Dear reviewer,
> > >
> > > We thank you for the supportive comments and positive assessment of the revision. We are pleased that the reviewer is fully satisfied with the revision.  We appreciate it very much!

---

### Official Review · Reviewer_Rvtn · 2022-10-25

**Confidence:** 3
**Correctness:** 3
**Technical Novelty And Significance:** 3
**Empirical Novelty And Significance:** 3
**Recommendation:** 8

**Clarity, Quality, Novelty And Reproducibility:**

The paper is clear and proposes a novel solution to an important problem; appears to be of high quality.

**Strength And Weaknesses:**

Strengths: The paper is very well-written and tackles an important problem concerning modern network datasets which typically have a large number of nodes. The proposed algorithm is based on a clear mathematical framework and supported by theoretical guarantees. It clearly performs very well in practice as illustrated by application to 4 real data sets.

Weaknesses: 1. The subsampling is achieved with a narrow objective of preserving the number of communities (rather than the content/quality of community structure which is possibly crucial to other global network summaries). A discussion on the latter and specifically the importance of the number of communities is currently not convincing.
2. If the  observed graph is assumed to be generated from a stochastic blockmodel, then the true number of communities is fixed and the problem makes sense. However, this is an assumption and may not be valid in practice. In such cases, estimating communities is simply a way to approximate the more general network generating process (e.g. Olhede and Wolfe, 2014) and a range of possible values for M are feasible (to lead to a reasonable approximation). This must be clarified in the discussion/introduction. 3. I would expect the performance of the proposed approach to additionally depend on the edge density of the network. This is an important feature as real networks often get sparser with increase in the number of nodes. This not discussed in the paper - neither in the theoretical results nor in the implementation. 4. Based on the OR curvature theory, are there any other network summaries concerning communities that the sub sampling procedure may possibly preserve?



**Summary Of The Paper:**

The paper proposes a graph sub-sampling procedure with the objective of preserving the number of communities M in the original (large) graph. This is based on Ollivier Ricci curvature (an extension of Ricci curvature for the setting of graphs). The paper notes that existing subsampling methods have a preference for sampling high degree nodes and as a consequence miss out on 'minority communities' (communities with a small number of low degree nodes) and hence underestimate the true number of communities. The algorithm is supported by theoretical guarantees and specifically that estimated M from the sampled subgraph (from their approach) converges to the true M of the large graph.

**Summary Of The Review:**

Overall, a good contribution which can possibly be improved to widen the scope of the problem currently addressed in the paper and with some additional discussion along the lines mentioned under weaknesses above.

---

> ### Author Response · Authors · 2022-11-19
> **Thank you for your comments (1/2)**
>
> We thank Reviewer Rvtn for the positive comments and for providing thoughtful feedback on our work. We address some of Reviewer Rvtn’s comments in our general response above, and we provide additional details on specific comments below.
>
> >  ***Comment:**  A discussion on the content/quality of community structure and specifically the importance of the number of communities is currently not convincing.*
>
> **Response:** Thank you for the comment. In response to your comment, we added the following discussion on the community structure and the importance of the number of communities in the introduction section.
>
> *"Network data often have natural communities, and identifying these communities helps answer vital questions in various fields (Rohe et al., 2011). For example, communities in social networks may represent groups of people who share a similar interest, and communities in protein-protein interaction networks could be regulatory modules of interacting proteins (Rohe et al., 2011). The theoretical properties of most community detection methods, such as consistency and asymptotic distributions, are built based on the assumption that $M$ is known (Ma et al., 2021). In addition, $M$ is usually required as an input for many community detection algorithms. However, in practice, we do not have the information of $M$, which significantly diminishes the usefulness of the aforementioned methods."*
>
> >  ***Comment:** If the observed graph is assumed to be generated from a stochastic block model, then the true number of communities is fixed and the problem makes sense. However, this is an assumption and may not be valid in practice.  This must be clarified in the discussion/introduction.*
>
> **Response:** Thank you for the suggestion. Please refer to our general responses **"Clarification on the setting"** to find our responses. In the revised manuscript, we added the discussion in the introduction section.
>
>
> >  ***Comment:**  I would expect the performance of the proposed approach to additionally depend on the edge density of the network. *
>
> **Response:** Thank you for the comment. Indeed, the performance of our method depends on the edge density from both theory and empirical results.
>
> On the one hand, Corollary 4.1.2 basically assumes that $\rho > \frac{log(\tilde{n})}{\tilde{n}}$, where $\tilde{n}$ is the number of sampled nodes, and $\rho$ is the edge density. When $\tilde{n}=100$, we require $\rho>0.046$, and when $\tilde{n}=1000$, we  only require $\rho>0.007$. As the number of sampled nodes becomes larger, the constraint imposed on the edge density becomes weaker.
>
>
> On the other hand, we include additional simulation studies to show the empirical performance of our method with different network densities. Specifically,  we use the same SBM setting in Section 5.1 to generate synthetic data, except that we let  $p_{in} = 0.8 \lambda$ and   $p_{out} = 0.1 \lambda$, where $\lambda \in \{0.2,0.4,\ldots, 1\}$ controls the edge density level. Here, we fix the sampling proportion as 10%; that is, we sample 90 nodes. Table R3 shows the edge density and average estimation error under different $\lambda$'s. As we can see, an increasing edge density comes with a lower error.
>
> In the revised manuscript, we added the above discussion on the relationship between the performance of our method and the edge density in Section C in the Appendix.
>
>
> Table R3:  Estimation error of the number of communities with different edge densities under the SBM model.
>
>  |$\lambda$|  0.2 | 0.4 | 0.6 | 0.8 | 1
> | :----  |    :----:   |    :----: | :----:   |    :----:   |   :----:
> |Edge density|  0.1018  | 0.2034 | 0.3053   |  0.4070  |  0.5085 |
> |Error | 2.7 | 2.1 | 1.8 | 1.8| 1.5 |

---

> ### Author Response · Authors · 2022-11-19
> **Thank you for your comments (2/2)**
>
>
> >  ***Comment:**  Based on the OR curvature theory, are there any other network summaries concerning communities that the sub-sampling procedure may possibly preserve?*
>
> **Response:**
> Thanks for the question. Yes! Theoretically, the OR curvature has been proven to be related to some network summaries concerning communities, such as the eigenvalues of the graph Laplace (Bauer et al., 2011) and the clustering coefficient (CC) (Jost & Liu, 2014).
>
> Empirically, we calculated the CC value of the full graph and that of the sampled subgraph for two real data: Facebook and Cora. We set the sampling proportion to 10\%. We replicated the sampling procedure 100 times and thus obtained 100 sampled subgraphs. Using these sampled subgraphs, we obtained the 95% confidence interval (CI) by calculating mean $\pm$ 1.96*standard deviation.
>
> For Facebook data, the CI of the CC value is [0.420, 0.537], and the CC value obtained from the full graph is  0.476. For Cora data, the CI is [0.234, 0.290], and the CC value of the full graph is 0.238. As we can see, the 95\% CI covers the CC value of the full graph for both two real data sets. These observations suggest that our sub-sampling procedure could preserve CC to some extent. The consistency theory on CC using our method is under investigation, and the results will be reported in future publications.
>
>
> References:
>
> [1] Frank Bauer, Jürgen Jost, and Shiping Liu. Ollivier-ricci curvature and the spectrum of the normalized graph laplace operator. *arXiv preprint*, arXiv:1105.3803, 2011.
>
> [2] Jürgen Jost and Shiping Liu. Ollivier’s ricci curvature, local clustering and curvature-dimension inequalities on graphs. *Discrete & Computational Geometry*, 51(2):300–322, 2014.

---

### Author Response · Authors · 2022-11-19
**Response to all reviewers (1/2)**

We thank the reviewers for their thoughtful and constructive review of our work. We were encouraged to hear the reviewers view the network sampling problem we present as important (Reviewer Rvtn) and well-motivated (Reviewer FvBB) and that they found our curvature-based sampling method as very nice and inspiring (Reviewer FvBB), novel (Reviewer yo3s, Rvtn), theoretically sound (Reviewer 5vqt), and clearly performs very well in practice (Reviewer Rvtn).

In response to feedback, we provide a general response to summarize all the major changes, individual responses to address each reviewer’s concerns, and an updated manuscript. In the revised manuscript, we highlight the changes in red.

### **1. Clarification on the setting**
In response to Reviewer  Rvtn and Reviewer yo3s,  we added the following statements to clarify this paper's setting in the third paragraph of the introduction.

*“In this paper, we focus on the setting where $M$ is a fixed model parameter, and there is a ground truth about $M$. This setting is widely used in many models, e.g., stochastic block model (SBM) and its variants, such as degree-corrected SBM (DCBM). The SBM family is arguably the most widely-used generative model for community detection from a theoretical perspective. Vaca-Ramírez & Peixoto (2022) performed a systematic analysis of the quality of fit of the SBM for 275 real networks and observed that “SBM is capable of providing an accurate description for the majority of networks considered.” Indeed, there are other settings where the number of communities is not fixed. For example, Olhede & Wolfe (2014) used SBM to approximate a nonparametric graphon model, under which case the number of communities is a hyperparameter and is not fixed. The latter hyperparameter case is beyond the scope of this paper.”*


### **2. Universality of  SBM**

Note that the setting where $M$ is a model parameter is widely used in the SBM family, including SBM  (Holland et al., 1983)  and its variants, such as degree-corrected SBM (DCBM) (Karrer & Newman, 2011). In what follows, we provide evidence to show that the SBM family is a sufficiently accurate representation of most real networks. This universality validates that the setting we considered in this paper is appropriate.

The SBM family model is arguably the most widely-used generative model to study community structures from a theoretical perspective. They also serve as generalizations of more fundamental random network models. The basic SBM has the Erdos-Rényi model as a special case, and likewise, the DCBM has the configuration model (Newman, 2010)   as a special case when there is a single group. Vaca-Ramírez & Peixoto (2022) performed a systematic analysis of the quality of fit of the SBM for 275 real networks, with the number of nodes ranging from 34 to 3,774,768. They observed that “SBM is capable of providing an accurate description for the majority of networks considered.”


Even though the theory of this paper is built under the SBM scenario, we empirically observe the superior performance of our method under both SBM and DCBM scenarios. In the future, we will extend the theory of our method to SBM variants.



### **3. Additional large real network**
Following the suggestion of Reviewer yo3s, we added one more real-world example, i.e., the PubMed citation network (Sen et al., 2008) with 19,717  nodes and  44,338 edges. The network density is only 0.0001; thus, PubMed is a very sparse network. The summary statistics of this network can be found in Table  A.4 in the updated Appendix. The estimation errors under different sampling proportions are presented in Table R1. As we can see, our method ORG-sub still outperforms other methods.


Table R1: The comparison of the estimation error of the PubMed dataset under different subsampling proportions.
 | Prop |  ORG-sub | MHRW | CSE  | FFS  | Snowball | DBN  |  RW  | MDRW |
|:--------:|:--------:|:----:|:-------:|:----:|:----:|:----:|:--------:|:----:|
| 0.5% | **0.20** | 0.30 | 0.30 | 0.50 | 1.00 | 0.70 |  0.30 | 1.90 |
| 2% | **0.00** | 0.30 | 0.80 | 0.40 | 0.20 | 0.70 | 1.20 | 1.80 |
| 5% | **0.30** | 0.55 | 0.55 | 1.40 | 1.65 | 0.65 | 2.20 | 2.75 |

---

> ### Author Response · Authors · 2022-11-19
> **Response to all reviewers (2/2)**
>
> ### **4. Computational Time**
> Following the suggestion of Reviewer FvBB and Reviewer  5vqt, in the updated appendix, we added Table A.5, which compares the computational time (s)  of estimating the number of communities using different sampling methods under different sampling proportions. Due to space limitations, in Table R2, we only show the computational time when the sampling proportion is 5\% for  PubMed and 10\% for other data. Since our method ORG-sub needs a preprocessing step, i.e., calculating  OR curvature, we also report the computational time of the curvature calculation in column "Ricci". From Table R2, we have three observations. First,  it is much cheaper to estimate using the sampling method than using the full graph. Second, ORG-sub has a comparable computation performance to other sampling methods. Third, the calculation of curvature is very fast, even for large network PubMed.
>
>
> Table R2:  Computational time of different methods. The second column means the computational time of using the full graph for estimation. The result of the full PubMed graph is not available (NA) due to prohibitive computational costs. "Ricci" means the computational time of  OR curvature.
>
>  | Dataset  |   Full   | Ricci | ORG-sub | MHRW | CSE  | FFS  | Snowball | DBN  |  RW  | MDRW |
> |:--------:|:--------:|:--------:|:-------:|:----:|:----:|:----:|:--------:|:----:|:----:|:----:|
> | Polblogs | **48.6** | 1.49 | 0.56 | 0.27 | 0.30 | 0.67 | 0.29 | 0.35 | 0.42 | 0.31 |
> | Polbooks | **1.88** | 0.29 | 0.10 | 0.05 | 0.07 | 0.08 | 0.10 | 0.05 | 0.07 | 0.04 |
> | Cora | **191.06** | 0.79 | 0.79 | 0.66 | 1.54 | 0.65 | 0.56 | 0.49 | 0.60 | 0.73 |
> | Facebook | **29.14** | 0.71 | 0.65 | 0.50 | 0.74 | 0.51 | 0.53 | 0.47 | 0.49 | 0.47|
> |PubMed | **NA** | 41.59 |8.92 | 8.61 | 43.36 | 8.94 | 8.24  | 6.62 | 8.83 | 7.85 |
>
> ### **5. Summary of changes in the revised manuscript**
>
> In the revised manuscript, we have included all aforementioned discussions and made the following changes.
> * Section 1: We have included a discussion about the setting of this paper and the universality discussion about SBM.
> * Section 2: We have followed the suggestion of Reviewer yo3s to improve the presentation.
> * Section 4: We have added a new theorem (i.e., Corollary 4.1.1)  to show our method's advantage over the random walk.
> * Section 5:  We have added (1) a real network, i.e., PubMed citation network, and  (2) a broader range of sampling proportions.
> * Section 6: We have included directions for future work.
> * Appendix: (1) We have added Table A.5  to show the computational time of real data of different methods under different sampling proportions. (2) Proof of the added theorem Corollary 4.1.1. (3) A section discussing related work on graph sampling.
> * Supplementary material: We provided code and data.
>
> We would like to thank all reviewers for their time and feedback again, and we hope that our changes adequately address all concerns.
>
> References:
>
> [1] Brian Karrer and Mark EJ Newman. Stochastic blockmodels and community structure in networks. *Physical review E*, 83(1):016107, 2011.
>
> [2] YX Rachel Wang and Peter J Bickel. Likelihood-based model selection for stochastic block models. *The Annals of Statistics*, 45(2):500–528, 2017.
>
> [3] Sofia C Olhede and Patrick J Wolfe. Network histograms and universality of blockmodel approximation. *Proceedings of the National Academy of Sciences*, 111(41):14722–14727, 2014.
>
> [4] Jing Lei. A goodness-of-fit test for stochastic block models. *The Annals of Statistics*, 44(1):401–424, 2016.
>
> [5] Peter J Bickel and Aiyou Chen. A nonparametric view of network models and newman–girvan and other modularities. *Proceedings of the National Academy of Sciences*, 106(50):21068–21073, 2009.
>
> [6] Tiago P Peixoto. Hierarchical block structures and high-resolution model selection in large networks. *Physical Review X*, 4(1):011047, 2014.
>
> [7] Fosdick, B. K., Larremore, D. B., Nishimura, J., & Ugander, J. (2018). Configuring random graph models with fixed degree sequences. *Siam Review*, 60(2), 315-355.
>
> [8] Felipe Vaca-Ramírez and Tiago P Peixoto. Systematic assessment of the quality of fit of the stochastic block model for empirical networks. *Physical Review E*, 105(5):054311, 2022.
>
> [9] Paul W Holland, Kathryn Blackmond Laskey, and Samuel Leinhardt. Stochastic blockmodels: First steps. *Social networks*, 5(2):109–137, 1983.
>
> [10] Brian Karrer and Mark EJ Newman. Stochastic blockmodels and community structure in networks. *Physical review E*, 83(1):016107, 2011.
>
> [11] M Newman. Networks: An introduction, new york, ny, usa: Oxford univ, 2010.
>
> [12] Shujie Ma, Liangjun Su, and Yichong Zhang. Determining the number of communities in degree-corrected stochastic block models. *Journal of machine learning research*, 22(69), 2021.
>
> [13] Prithviraj Sen, Galileo Mark Namata, Mustafa Bilgic, Lise Getoor, Brian Gallagher, and Tina Eliassi-Rad. Collective classification in network data. *AI Magazine*, 29(3):93–106, 2008.

---

### Decision · Program_Chairs · 2023-01-20

**Decision:**

Accept: poster

**Justification For Why Not Higher Score:**

The idea of using Ricci curvature for sampling is interesting. The paper presents both theory and experiments to support their claims.

**Justification For Why Not Lower Score:**

The experimental results are not particularly convincing, expecially because the analyzed dataset are quite small

**Metareview: Summary, Strengths And Weaknesses:**

The authors propose a new method for sampling subgraphs to obtain a good representation of the network. The main idea behind the results is that classic subsampling strategies(based on node degrees or random walks) fail to capture the structure of small network. To cope with this limitation they present a new sampling strategy based on Ricci curvature.

The authors show the effectiveness of their method both theoretically(by analyzing the SBM model) and empirically.

Overall, the ideas presented in the paper are interesting so it will be a nice contribution to the conference program.

**Note From Pc:**

if the above contains the word "oral" or "spotlight" please see: "oral" presentation means -> notable-top-5% and "spotlight" means -> notable-top-25%. As stated in our emails, we are disassociating presentation type from AC recommendations